# RobustVLA: Robustness-Aware Reinforcement Post-Training for Vision-Language-Action Models

## Abstract

Vision-Language-Action (VLA) models have recently emerged as powerful general-purpose policies for robotic manipulation, benefiting from large-scale multi-modal pre-training. However, they often fail to generalize reliably in out-of-distribution deployments, where unavoidable disturbances such as observation noise, sensor errors, or actuation perturbations become prevalent. While recent Reinforcement Learning (RL)-based post-training provides a practical means to adapt pre-trained VLA models, existing methods mainly emphasize reward maximization and overlook robustness to environmental uncertainty. In this work, we introduce **RobustVLA**, a lightweight online RL post-training method designed to explicitly enhance the resilience of VLA models. Through a systematic robustness analysis, we identify two key regularizations: Jacobian regularization, which mitigates sensitivity to observation noise, and smoothness regularization, which stabilizes policies under action perturbations. Extensive experiments across diverse robotic environments demonstrate that RobustVLA significantly outperforms prior state-of-the-art methods in robustness and reliability. Our results highlight the importance of principled robustness-aware RL post-training as a key step toward improving the reliability and robustness of VLA models.

## 1 Introduction

Vision-Language-Action (VLA) models have emerged as a powerful paradigm for general-purpose robotic manipulation, with representative examples including RT-1 (Brohan et al., 2022), RT-2 (Zitkovich et al., 2023), OpenVLA (Kim et al., 2024), and $\pi_0$ (Black et al., 2024). These VLA models demonstrate the feasibility of learning universal robotic policies from large-scale multimodal datasets and have achieved remarkable performance across a variety of manipulation tasks. Despite these advances, recent studies (Xing et al., 2025; Shi et al., 2025) reveal that the generalization ability of pre-trained VLA models remains limited when deployed in Out-of-Distribution (OOD) scenarios. The OOD scenarios in this work primarily refer to variations in the distribution of perception and execution levels. These variations are caused by unseen visual conditions (lighting, occlusion, camera changes, etc.) and execution disturbances (action noise). One key reason is that VLA models may rely on spurious correlations between irrelevant observation features and actions, rather than capturing the true causal relationship required for robust generalization (Xing et al., 2025).

A natural solution for OOD tasks is Supervised Fine-Tuning (SFT) with task-specific data. However, SFT approaches based on imitation learning rely heavily on costly, high-quality human demonstrations. When only a limited number of demonstrations are available, their effectiveness drops sharply. In contrast, online Reinforcement Learning (RL) provides an appealing alternative by enabling autonomous data collection and policy refinement in deployment (Lu et al., 2025; Liu et al., 2025; Shu et al., 2025; Chen et al., 2025c; Tan et al., 2025a). However, while recent online RL-based post-training methods enable autonomous adaptation without requiring large amounts of expert demonstrations, they are not inherently designed to guarantee robustness against environmental perturbations. General RL fine-tuning focuses on maximizing task-specific reward signals in nominal environments, but does not explicitly regularize the sensitivity of the VLA model to environmental noise. Such optimized models are often over-adapted to the specific dynamics of the post-training environment and become brittle to even slight perturbations. This limitation highlights the need for

principled approaches that not only optimize for task performance, but also explicitly constrain the sensitivity of the model's decision process to perturbations. Such robustness-oriented post-training is crucial for ensuring reliable deployment of VLA models in real-world environments.

Environmental perturbations can arise from multiple sources and affect different stages of the model-environment interaction (Gu et al., 2025). Currently, we focus on two fundamental types: observation perturbations and action perturbations. Observation perturbations reflect inaccuracies in perception, caused by sensor noise, latency, or imperfect state estimation, which lead to mismatches between the perceived and actual environment state. Action perturbations, on the other hand, arise from actuation errors, implementation inaccuracies, or hardware-level disturbances, which cause deviations between the intended and executed actions. These two perturbations, which are ubiquitous in real-world robotic systems, can be effectively modeled as random noise applied to the sequential decision-making process (Duan et al., 2016). Furthermore, if not properly addressed, these perturbations can amplify over time and significantly degrade model performance. Thus, a significant gap remains: *ensuring the robustness of post-training to environmental perturbations, so that the resulting VLA model is stable and reliable in autonomous interactions.*

To this end, we propose **RobustVLA**, a concise and effective online RL post-training method designed to enhance the robustness of pre-trained VLA models under environmental perturbations. Our method is grounded in the robustness analysis of performance deviations induced by perturbations. We first establish explicit upper bounds on the robustness gap in the presence of observation and action perturbations, respectively. We then extend this analysis to the case where both types of perturbations coexist. Our findings highlight that the robustness gap under observation perturbations is governed by the Jacobian sensitivity of the VLA model, whereas under action perturbations it is controlled by the smoothness of the VLA model update. This theoretical insight naturally motivates the introduction of Jacobian and smoothness regularization into the online RL fine-tuning process, which together constrain the worst-case robustness gap and promote robust decision-making. The main contributions of this work are summarized as follows:

- We introduce RobustVLA, a lightweight online RL post-training method that enhances the robustness of VLA models against environmental perturbations.

- We conduct three robustness analyses to performance deviations caused by observation and action perturbations, which induce explicit VLA model regularization optimization terms.

- Extensive experiments demonstrate that RobustVLA exhibits superior resistance to environmental uncertainties and perturbations compared to state-of-the-art VLA baselines, resulting in more reliable and stable behavior.

## 2 RELATED WORK

**RL for VLA Models.** Recently, a growing body of research has explored RL as a mechanism to adapt VLA models to complex embodied tasks. The first research direction emphasizes offline RL training, where large-scale policy optimization can be performed without expensive online interactions. Previous work proposed the Q-Transformer (Chebotar et al., 2023), introducing autoregressive offline Q-learning with a Transformer. Subsequently, in the work by ReinboT (Zhang et al., 2025c), researchers proposed an end-to-end VLA model that incorporates the RL principle of maximizing cumulative return. Further work has proposed an offline RL post-training algorithm, ARFM (Zhang et al., 2025b), for VLA flow models to boost the performance of VLA models on downstream tasks. Moreover, some work has focused on online RL fine-tuning of VLA models. FLaRe (Hu et al., 2024) employs large-scale domain randomized RL fine-tuning, and PA-RL (Mark et al., 2024) unifies offline and online RL across policy categories. IRe-VLA (Guo et al., 2025), RIPT-VLA (Tan et al., 2025a), and VLA-RL (Lu et al., 2025) are specifically designed to stabilize or extend online RL training of VLA models. Furthermore, optimization advances such as TGRPO (Chen et al., 2025c) and RFTF (Shu et al., 2025) improve the efficiency of fine-tuning in trajectory or sparse reward settings. RLDG (Xu et al., 2024) refines task-specific RL into a general VLA, ConRFT (Chen et al., 2025b) combines offline Q-learning with consistency-based online fine-tuning, and ReWiND (Zhang et al., 2025d) introduces a language-guided reward model for fast task adaptation without the need for new demonstrations. Moreover, several works have focused on training value functions for visual-language manipulation tasks to align VLA models with downstream tasks, including Bellman-Guided Retrials (Du et al., 2024), V-GPS (Nakamoto et al., 2024),

and Hume (Song et al., 2025) However, previous work has often neglected the robustness and reliability of robotic visual-language manipulation under environmental perturbations and distribution changes, which is precisely the focus of our work. More related works are in Appendix A.1.

**Robust RL.** Robust RL generally studies how to ensure stability and generalization under distribution shifts, noisy observations, and adversarial perturbations. Classical approaches leverage robust MDP formulations (Iyengar, 2005; Wiesemann et al., 2013) and adversarial training (Pinto et al., 2017). Modern deep RL has introduced diverse techniques, including domain randomization for sim-to-real transfer (Tobin et al., 2017; Van et al., 2025) and methods for offline and curriculum learning (Dennis et al., 2020; Wu et al., 2024). Recently, the field has advanced towards theoretically-grounded distributionally robust optimization (Li et al., 2024; Cui et al., 2025; He et al.) with provable finite-sample guarantees (Ghosh et al., 2025; Roch et al., 2025). However, the online setting where an agent learns via direct interaction is less studied. A key challenge, unaddressed by previous work that often assumes access to exploratory policies (Wang & Zou, 2021; Badrinath & Kalathil, 2021), is to handle exploration and robustness simultaneously. Crucially, this focus on test-time robustness to environmental shifts distinguishes distributionally robust RL from corruption-robust RL, which handles corrupted training data (Lykouris et al., 2021; Wei et al., 2022; Zhang et al., 2022; Ye et al., 2024; 2023). Different from these, our work focuses on the post-training of VLA models for general task settings, aiming to simultaneously address model adaptability and stability issues within a unified model framework.

## 3 PRELIMINARIES

**RL post-training of VLA model.** We model the task as a Markov Decision Process (MDP) (Sutton et al., 1998). The VLA model $\pi_\theta$ maps a natural language instruction $g$ and a sequence of observations $o_{1:t}$ and previous actions $a_{1:t-1}$ to a probability distribution over the current action $a_t$. Each episode is initialized with the context $\mathbf{c} = (g, o_1)$. The state $s_t$ is represented as $[g, o_{1:t}, a_{1:t-1}]$. At each time step $t$, $\pi_\theta$ samples an action from the model distribution: $a_t \sim \pi_\theta(\cdot \mid g, o_{1:t}, a_{1:t-1})$. The environment $\mathcal{E}$ transitions to the next state $s_{t+1} = [g, o_{1:t+1}, a_{1:t}]$. The environment $\mathcal{E}$ also returns a binary reward $r_t = 1$ if the task goal is achieved, and $r_t = 0$ otherwise. The goal of $\pi_\theta$ is to maximize the undiscounted task reward: $J(\pi_\theta) = \mathbb{E}_{\tau \sim \pi_\theta}[\sum_{t=1}^{H} r_t]$, where $H$ is the rollout horizon. To optimize this objective, policy gradient (Sutton et al., 1999) is a simple and efficient method: $\nabla_\theta J(\pi_\theta) = \mathbb{E}_{\mathbf{a} \sim \pi_\theta}[\nabla_\theta \log \pi_\theta(\mathbf{a} \mid \mathbf{c}) A(\mathbf{c}, \mathbf{a})]$, where $A(\mathbf{c}, \mathbf{a})$ is the advantage function, which indicates how good the action $\mathbf{a}$ is compared to the baseline. However, the accurate estimation of $A(\mathbf{c}, \mathbf{a})$ is tricky, especially in the post-training phase of VLA models. To alleviate this issue, recent work (Chen et al., 2025a) proposed a critic-free optimization framework. Specifically, for each sampling context $\mathbf{c}$, we can draw $K$ rollouts $\{\mathbf{a}_k \sim \pi_\psi(\cdot \mid \mathbf{c})\}_{k=1}^{K}$ under a sampling model $\pi_\psi$. Each rollout receives a reward $r_k = r(\mathbf{c}, \mathbf{a}_k)$. The leave-one-out baseline for rollout $k$ is calculated by averaging the other rewards (Kool et al., 2018): $A_k = r_k - \frac{1}{K-1} \sum_{j \neq k} r_j$. This allows us to bypass the cumbersome task of learning a value function and efficiently compute a stable advantage signal. Furthermore, to utilize the collected rollouts $\{(\mathbf{c}_k, \mathbf{a}_k, A_k)\}$ to update $\pi_\theta$, we compute the importance ratio: $\eta_k = \pi_\theta(\mathbf{a}_k \mid \mathbf{c}_k) / \pi_\psi(\mathbf{a}_k \mid \mathbf{c}_k)$, where $\pi_\theta$ is the current update model and $\pi_\psi$ is a sampling model. We then optimize $\pi_\theta$ utilizing the Proximal Policy Optimization (PPO) objective (Schulman et al., 2017):

$$\mathcal{L}_{\text{PPO}} = -\min\left(\eta_i A_i, \text{clip}\left(\eta_i, 1 - \epsilon, 1 + \epsilon\right) A_i\right), \tag{1}$$

where $\epsilon$ is an update threshold. This objective encourages rollouts with positive advantages while preventing $\pi_\theta$ from deviating significantly from the sampling model $\pi_\psi$.

**Environmental Perturbations During RL Post-Training.** We formalize the robust post-training problem of VLA model $\pi_\theta$ under environmental perturbations. At each time step $t$ in MDP, the $\pi_\theta$ receives a perturbed observation $\tilde{s}_t$ of the true state $s_t$, with deviation bounded by $\|\tilde{s}_t - s_t\| \leq \epsilon_s$. The action selected by the $\pi_\theta$ may be corrupted by execution noise, modeled as Gaussian perturbations: $a_t = \pi_t(\tilde{s}_t) + \xi_t$, where $\xi_t \sim \mathcal{N}(0, \sigma^2 I_d)$. The environment evolves according to a Lipschitz continuous transition function $f$, ensuring that small deviations in states or actions yield bounded deviations in subsequent states. The environmental dynamics $f(s, a)$ and the reward function $r(s, a)$ are Lipschitz continuous and are constants $L_f$ and $L_r$, respectively. Finally, we denote by $\epsilon_{\text{offline}}$ the discrepancy between the initial VLA model $\pi_{init}$, obtained via imitation learning, and the expert model $\pi^*$. This discrepancy provides the starting error baseline for the online post-training process.

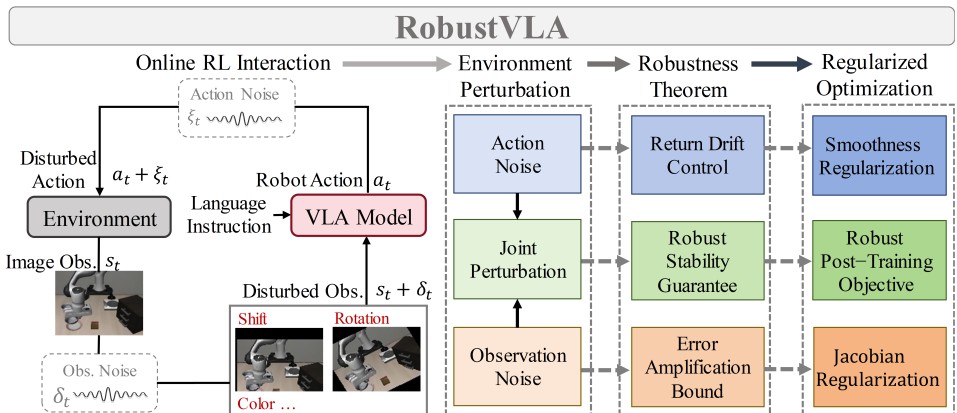

Figure 1: The proposed RobustVLA method. Due to the presence of environmental uncertainty during online RL interactions, we consider observation noise (sensor/camera corruptions) and action noise (Gaussian actuation errors), and their joint effect. Moreover, we conduct robustness theoretical analysis based on these three aspects, establishing error amplification bounds, return drift control, and robust stability guarantees. Finally, we derive regularized optimization objectives, including the model Jacobian and action smoothing regularization, as well as the robust RL post-training objective.

# 4 METHODOLOGY

In this work, we propose a novel robust online RL post-training method, RobustVLA, for VLA models under environmental perturbations (Fig. 1). We first analyze the robustness bounds of the VLA model under random environmental perturbations. We then discuss how to derive post-training objectives based on these analyses and finally propose a concise online RL fine-tuning algorithm.

## 4.1 ROBUSTNESS ANALYSIS TO ENVIRONMENTAL PERTURBATIONS

In the perturbed online RL post-training of the VLA model, we observed two main sources of robustness degradation. First, these models are highly sensitive to small changes in visual input: slight shifts in lighting, occlusions, or camera noise can cause disproportionately large changes in the model's latent representation and therefore in its actions. This motivates explicitly constraining how sharply the model reacts to observation changes. Second, when the model is updated online, its behavior can change abruptly from one iteration to the next. Even small gradient steps may lead to large jumps in the produced actions, which become unstable when combined with the stochasticity of realistic execution. To stabilize this process, we also need to control how quickly the model is allowed to evolve during post-training. Therefore, in this section, we theoretically quantify how different types of perturbations affect the robustness of the VLA model.

**Theorem 1** (**Error Amplification Bound**). *Assume perturbed observations $\tilde{s}_t = s_t + \delta_s^t$ with $\|\delta_s^t\| \le \epsilon_s$, and bounded Jacobian $\|\nabla_s \pi_t(s)\| \le \lambda$. Then:*

$$\mathbb{E}\big[J(\pi^*) - J(\pi)\big] \le \mathcal{O}(H L_r L_f^H) \cdot (\epsilon_{offline} + \lambda \epsilon_s).$$

Here the factor $L_f^H$ corresponds to the worst-case geometric amplification under $L_f$-Lipschitz dynamics; when $L_f < 1$ the bound becomes strictly smaller. Theorem 1 indicates that the robustness gap under observation perturbations is governed by the product $\lambda \epsilon_s$, where $\lambda$ measures the VLA model's local sensitivity and $\epsilon_s$ quantifies the observation noise level. Conversely, enforcing Jacobian regularization to shrink $\lambda$ directly reduces the error propagation term $\lambda \epsilon_s$, thereby bounding $J(\pi^*) - J(\pi)$ more tightly.

**Theorem 2** (**Return Drift Control**). *Assume actions are perturbed as $a_t = \pi_t(s_t) + \xi_t$ with $\xi_t \sim \mathcal{N}(0, \sigma^2 I_d)$, and VLA models satisfy $\|\pi_i - \pi_{i-1}\|_\infty \le \delta_i$. Then:*

$$\mathbb{E}\big[J(\pi^*) - J(\pi)\big] \le \mathcal{O}(H L_r L_f^H) \cdot \Big(\epsilon_{\text{offline}} + \sum_{i=1}^{N} \delta_i + \sigma\sqrt{d}\Big).$$

Here, $N$ represents the number of times the model parameters are updated. Theorem 2 reveals that the return gap under action perturbations has two additive drivers: **1)** the cumulative VLA model drift $\sum_{i=1}^{N} \delta_i$, which reflects the nonstationarity of online updates, and **2)** the stochastic execution noise term $\sigma\sqrt{d}$, which is irreducible and scales with action dimension $d$. The presence of $\sum_{i=1}^{N} \delta_i$ shows that if VLA model updates are too aggressive (large $\delta_i$), then even in the absence of large execution noise ($\sigma \approx 0$), the compounding drift will destabilize rollouts and inflate $J(\pi^*) - J(\pi)$. This means smoothness regularization $--$ enforcing $\delta_i$ to be small $--$ is essential to decouple online learning progress from robustness degradation, ensuring that long-horizon performance does not deteriorate due to accumulated VLA model shifts.

**Theorem 3** (**Robust Stability Guarantee**). *Under both observation and action perturbations, and both Jacobian and smooth regularizations, the return gap satisfies:*

$$\mathbb{E}\big[J(\pi^*) - J(\pi)\big] \leq \mathcal{O}(HL_r L_f^H) \cdot \left( \epsilon_{\text{offline}} + \sum_{i=1}^{N} \delta_i + \lambda\epsilon_s + \sigma\sqrt{d} \right).$$

Theorem 3 shows that when both observation and action perturbations are present, the return gap depends jointly on the observation–sensitivity term $\lambda\epsilon_s$, and the accumulated update drift $\sum_{i=1}^{N} \delta_i$. Unlike the single-perturbation cases, here the deviation caused by noisy observations does not remain localized: it propagates through the dynamics, influences later states, and interacts with execution noise and model drift across the entire rollout. This nested feedback structure implies that robustness cannot be achieved by controlling only one source of error. Instead, Jacobian regularization (which reduces the instantaneous sensitivity $\lambda$) and smoothness regularization (which limits update-induced drift $\delta_i$) must be applied jointly to prevent compounding error growth over long horizons. This result highlights that robust VLA post-training fundamentally requires a dual regularization strategy; ignoring either perturbation channel will allow errors to accumulate through the dynamics and ultimately destabilize the VLA model. More robustness analysis is in the Appendix A.2.

## 4.2 MODEL ONLINE REGULARIZATION OPTIMIZATION OBJECTIVE

The robustness analysis in Section 4.1 suggests two complementary levers: **1)** Jacobian regularization to suppress error amplification from observation noise, achieved by penalizing the sensitivity of the model with respect to its input states. **2)** Smoothness regularization to stabilize online updates against action perturbations, achieved by penalizing deviations between consecutive models. In our implementation, Jacobian regularization is realized by computing the gradient of the log-action probability with respect to the observation input: $\nabla_s \log \pi_\theta(a|s)$. Specifically, we compute the squared norm of this gradient across a batch of observations, average the result, and then apply a clamp operation to prevent excessively large penalties:

$$\mathcal{R}_{\text{Jac}}(\theta) = \mathbb{E}_{(s,a)\sim\mathcal{D}} \left[ \min\left\{ \left\| \nabla_s \log \pi_\theta(a|s) \right\|_2^2, G_{max} \right\} \right], \tag{2}$$

where $G_{max}$ is the gradient clamp parameters. The theoretical justification for applying Jacobian regularization on the log-probability rather than the raw probability distribution comes from a norm comparison argument. Since the model $\pi_\theta(a|s)$ is a probability distribution, which induces the following relationship: $\left\| \nabla_s \pi_\theta(a|s) \right\|_F \leq \left\| \nabla_s \log \pi_\theta(a|s) \right\|_F$. This is because $\nabla_s \log \pi_\theta(a|s) = \nabla_s \pi_\theta(a|s)/\pi_\theta(a|s)$, and since $0 < \pi_\theta(a|s) \leq 1$, dividing by $\pi_\theta(a|s)$ only enlarges the magnitude of the gradient. Consequently, any control on $\left\| \nabla_s \log \pi_\theta(a|s) \right\|_F$ automatically controls $\left\| \nabla_s \pi_\theta(a|s) \right\|_F$ as well, but in a stronger sense. This directly penalizes large sensitivity coefficients, aligning with the bound in Theorem 1.

On the other hand, Theorem 2 states that the return gap can be bounded in terms of the worst-case deviation $\delta_i = \sup_{s\in\mathcal{S}} \| \pi_\theta(\cdot|s) - \pi_{\theta^-}(\cdot|s) \|_\infty$, which captures the maximum discrepancy between the new and old models across the entire state space. Although this characterization is theoretically precise, computing or constraining $\delta_i$ directly is infeasible in practice, especially in continuous action domains. To operationalize this idea, we introduce easy-to-handle alternatives. First, the PPO objective already constrains the average distributional shift through a KL divergence penalty: $\mathbb{E}_{s\sim\mathcal{D}}\Big[ D_{\text{KL}}\big( \pi_\theta(\cdot|s) \,\|\, \pi_{\theta^-}(\cdot|s) \big) \Big] \leq \epsilon$. By Pinsker's inequality, this implies $\| \pi_\theta(\cdot|s) - \pi_{\theta^-}(\cdot|s) \|_1 \leq \sqrt{2D_{\text{KL}}\big( \pi_\theta(\cdot|s)\|\pi_{\theta^-}(\cdot|s) \big)}$, thus providing an average-case upper bound

---

**Algorithm 1** Robust online RL Post-Training for VLA Models

---

**Input:** Pretrained VLA model $\pi_\theta$, rollout buffer $\mathcal{D}_{\text{rollout}}$, Jacobian weight $\alpha$, smoothness weight $\beta$, gradient clamp $G_{\max}$, noise range $(\epsilon_{\min}, \epsilon_{\max})$, success rate thresholds $(\tau_{\text{low}}, \tau_{\text{high}})$.

---

1: Initialize reference model $\pi_{\theta^-} \leftarrow \pi_\theta$, noise level $\epsilon \leftarrow \epsilon_{\min}$, success moving average $p_{MA} \leftarrow 0$
2: **for** $m \leftarrow 1$ **to** $M$ **do**
3:     Collect trajectories $\mathcal{D}_{\text{rollout}}$ with observation/action noise $\epsilon$
4:     **for** $n \leftarrow 1$ **to** $N$ **do**
5:         Sample minibatch $(s_i, a_i, A_i) \sim \mathcal{D}_{\text{rollout}}$
6:         Jacobian penalty: $\mathcal{R}_{\text{Jac}} = \mathbb{E}_i\big[\min(\|\nabla_{s_i} \log \pi_\theta(a_i|s_i)\|_2^2, G_{\max})\big]$
7:         Action-smooth penalty: $\mathcal{R}_{\text{Smooth}} = \mathbb{E}_i[\|\mu_\theta(s_i) - \mu_{\theta^-}(s_i)\|_2^2]$
8:         Robust objective: $\mathcal{L}_{\text{RobustVLA}} = \mathcal{L}_{\text{PPO}} + \alpha\mathcal{R}_{\text{Jac}} + \beta\mathcal{R}_{\text{Smooth}}$
9:         Update $\theta \leftarrow \theta - \eta\nabla_\theta\mathcal{L}_{\text{RobustVLA}}$
10:     **end for**
11:     Update reference $\pi_{\theta^-} \leftarrow \pi_\theta$
12:     **if** $m \% I_{\text{interval}} = 0$ **then**
13:         Evaluate model $\pi_\theta$, obtain success rate $p$
14:         Update moving average $p_{MA} \leftarrow \gamma p + (1 - \gamma)p_{MA}$
15:         **if** $p_{MA} > \tau_{\text{high}}$: $\epsilon \leftarrow \min(\epsilon + \Delta, \epsilon_{\max})$
16:         **else if** $p_{MA} < \tau_{\text{low}}$: $\epsilon \leftarrow \max(\epsilon - \Delta, \epsilon_{\min})$
17:     **end if**
18: **end for**
19: **Return** finetuned robust VLA model $\pi_\theta$

---

on model deviation. However, the KL penalty tends to be sensitive to variance changes and does not necessarily prevent large shifts in the mean action outputs. To complement this, we further introduce a smoothness regularizer that explicitly penalizes deviations in the model means:

$$\mathcal{R}_{\text{Smooth}}(\theta) = \mathbb{E}_{s\sim\mathcal{D}}\left[\|\mu_\theta(s) - \mu_{\theta^-}(s)\|_2^2\right], \tag{3}$$

where $\mu_\theta(s)$ denotes the mean action. This regularizer directly bounds the displacement of the mean actions, which in turn controls a surrogate of $\delta_i$ in expectation. Intuitively, while the KL constraint ensures distribution-level stability, the $\mathcal{R}_{\text{Smooth}}$ enforces gradual updates in the action means, thereby preventing erratic model shifts that may otherwise amplify error accumulation. This dual mechanism $--$ average KL regularization together with mean-level $L_2$ smoothing $--$ serves as a practical surrogate for bounding $\delta_i$, thus aligning the algorithmic implementation with the theoretical insights of Theorem 2. Therefore, taking these regularization terms into consideration, we obtain a robust online RL optimization objective:

$$\mathcal{L}_{\text{RobustVLA}}(\theta) = \mathcal{L}_{\text{PPO}}(\theta) + \alpha\,\mathcal{R}_{\text{Jac}}(\theta) + \beta\,\mathcal{R}_{\text{Smooth}}(\theta), \tag{4}$$

where $\alpha$ and $\beta$ are hyperparameters controlling the strength of each robustness component. This formulation preserves the general PPO advantage-based policy improvement while explicitly constraining the two key robustness channels identified in our analysis: sensitivity to observation perturbations ($\alpha$ term) and stability under action noise and online updates ($\beta$ term). As a result, $\mathcal{L}_{\text{RobustVLA}}$ naturally operationalizes the theoretical bounds, guiding the learning process toward VLA models that are both performant and robust under environmental perturbations.

### 4.3 ROBUST POST-TRAINING ALGORITHM IMPLEMENTATION

Based on the constructed robust RL optimization objective (Equ. 4), we can design a lightweight VLA model post-training algorithm as shown in Algo. 1. The algorithm retains the rollout-collection and model-update structure of PPO, but crucially incorporates three key components beyond the standard objective. First, the Jacobian penalty $\mathcal{R}_{\text{Jac}}$ suppresses the sensitivity of the model to input perturbations by penalizing large gradients of $\log \pi_\theta(a|s)$ with respect to observations. Second, the action-smooth penalty $\mathcal{R}_{\text{Smooth}}$ constrains the change of action distributions between consecutive model updates, thereby ensuring temporal consistency. Finally, to further enhance robustness in practice, we adopt a curriculum-based adaptive noise scheduling mechanism: the level of injected

observation and action noise is gradually increased or decreased based on the model's smoothed success rate. This strategy prevents premature destabilization during training, while gradually exposing the model to stronger perturbations as competence improves. Together, these design choices ensure that the model update not only maximizes advantages under the PPO surrogate loss, but also maintains robustness against both observation and action perturbations, thereby grounding the theoretical analysis into a practical and implementable training scheme.

## 5 EXPERIMENTAL EVALUATION

In this section, we conduct extensive experiments in various scenarios to evaluate the effectiveness of the proposed RobustVLA. Specifically, we aim to examine the three key questions: **1)** Does RobustVLA outperform state-of-the-art offline and online baseline methods in perturbed environment settings? **2)** How does RobustVLA perform in online transfer learning under perturbed scenarios? **3)** How do the core components of RobustVLA affect the overall performance?

### 5.1 PERFORMANCE EVALUATION UNDER ENVIRONMENTAL PERTURBATIONS

**Experimental setup.** To comprehensively evaluate the performance of the RobustVLA, we construct a robust testing benchmark based on the LIBERO (Liu et al., 2023) simulation platform that incorporates environmental uncertainty and perturbations, as shown in Fig. 2. The original LIBERO benchmark is a long-horizon robotic learning suite consisting of **Objects**, **Long**, **Spatial**, and **Goal**. We impose five observation perturbations and three action perturbations during the online interaction between the VLA model and the environment. The five observation perturbations are Image **Shift**, Image **Rotation**, **Color** Jitter, Image **Occlusions**, and Image **Erasing**. The three action

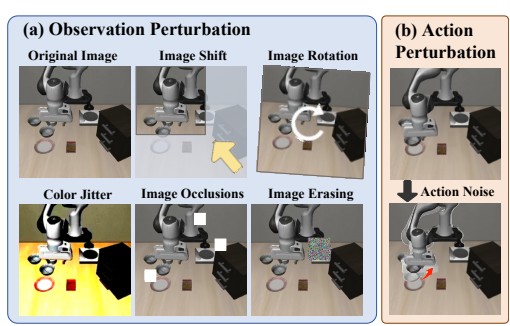

Figure 2: Robust VLA benchmarks based on the LIBERO include two types: **a)** observation perturbation and **b)** action perturbation.

perturbations are zero-mean Gaussian action noise with standard deviations of **0.1**, **0.2**, and **0.3**. Details of the observation perturbations are in Appendix A.4. For model evaluation, we follow the common scheme of previous work (Tan et al., 2025a), where a single VLA model is deployed to all tasks in the suite, and the deployment is performed on 50 held-out test contexts for each task.

**Baselines.** Our proposed algorithm comes in two versions: one without curriculum learning (**RobustVLA**) and one with curriculum learning (**RobustVLA-C**). For a thorough comparison, three categories of state-of-the-art baseline models are considered. The first category is Offline Imitation Learning (**Offline IL**) models, which are trained solely on static expert datasets. These include: **1)** GEVRM (Zhang et al., 2025a): This model uses prototype contrastive learning to resist observation perturbations. **2)** OpenVLA (Kim et al., 2024), which represents robot actions as discrete token sequences and predicts them one by one in sequence. **3)** OpenVLA-OFT (Kim et al., 2025), which follows the OpenVLA modeling philosophy and further proposes improvements such as parallel decoding. **4)** $\pi_0$ (Black et al., 2024), a VLA flow model that effectively models multi-modal distributions. The second category is Offline Reinforcement Learning (**Offline RL**) models, which apply RL algorithms to offline datasets to more fully capture the data quality distribution. These include: **5)** RWR (Peters & Schaal, 2007), which optimizes the model by performing reward-weighted regression on samples. **6)** ReinboT (Zhang et al., 2025c), which guides VLA action generation by predicting densely maximized future returns. **7)** ARFM (Zhang et al., 2025b), a fine-tuning algorithm that adaptively adjusts the weights of a flow-matching RL loss. The final category is Online Reinforcement Learning (**Online RL**) models, which fine-tune a pre-trained VLA model by autonomously interacting with the environment. This includes **8)** RIPT-VLA (Tan et al., 2025a), a critic-free fine-tuning algorithm that uses only sparse binary success rewards. The implementation details of our method and the reproduction of the baseline algorithm are shown in Appendix A.5.

Table 1: Average Success Rate (SR) (%) under five observation perturbations. The best results are highlighted in bold, and the second-best results are underlined.

| Model Type | Models | Observation Perturbation | | | | | Avg. |
|---|---|---|---|---|---|---|---|
| | | Shift | Rotation | Color | Occlusions | Erasing | |
| Offline IL | $\pi_0$ | 31.2 | 49.5 | 78.5 | 88.3 | 85.2 | 66.5 |
| | GEVRM | **46.4** | 57.9 | 56.4 | 59.8 | 63.3 | 56.8 |
| | OpenVLA | 25.4 | 41.6 | 48.5 | 75.5 | 73.6 | 47.9 |
| | OpenVLA-OFT | 40.2 | 75.1 | **95.9** | 95.2 | 96.4 | 80.6 |
| Offline RL | RWR | 16.3 | 37.1 | 74.8 | 88.2 | 83.6 | 60.0 |
| | ARFM | 11.9 | 45.0 | 75.5 | 88.0 | 84.0 | 60.9 |
| | ReinboT | 19.8 | 37.1 | 74.4 | 89.1 | 84.9 | 61.0 |
| Online RL | RIPT-VLA | 39.5 | 79.0 | 95.1 | 95.0 | 95.6 | 80.8 |
| | **RobustVLA** | 42.1 | 82.6 | 95.6 | **95.7** | **96.9** | **82.5** |
| | **RobustVLA-C** | 41.1 | **82.8** | 92.3 | 95.5 | 96.5 | 82.2 |

Table 2: Average SR (%) under action perturbations (noise level = 0.1, 0.2, 0.3).

| Models | Action Perturbation | | | Avg. |
|---|---|---|---|---|
| | 0.1 | 0.2 | 0.3 | |
| $\pi_0$ | 77.5 | 42.5 | 18.4 | 46.1 |
| OpenVLA | 53.6 | 39.2 | 11.7 | 34.8 |
| OpenVLA-OFT | 87.4 | 52.3 | 20.7 | 53.5 |
| RWR | 80.6 | 44.8 | 20.8 | 48.7 |
| ARFM | 81.1 | 48.2 | 21.1 | 50.1 |
| ReinboT | 80.3 | 47.0 | 21.2 | 49.5 |
| RIPT-VLA | 84.7 | 48.1 | 18.8 | 50.5 |
| **RobustVLA** | 88.5 | 53.1 | **22.7** | **54.8** |
| **RobustVLA-C** | **88.9** | **53.3** | 21.8 | 54.7 |

Table 3: Average SR (%) of the four LIBERO suites under joint perturbations.

| Models | Joint Perturbation | | | | Avg. |
|---|---|---|---|---|---|
| | Spatial | Object | Goal | Long | |
| $\pi_0$ | 59.0 | 33.2 | 56.2 | 35.0 | 45.9 |
| GEVRM | 78.6 | 42.0 | 67.8 | 33.4 | 55.5 |
| OpenVLA | 20.4 | 1.8 | 21.0 | 0.0 | 10.8 |
| OpenVLA-OFT | 84.2 | 71.2 | 81.6 | 40.8 | 69.5 |
| RWR | 60.4 | 39.6 | 53.6 | 38 | 47.9 |
| ARFM | 58.0 | 36.6 | 54.6 | 37.2 | 46.6 |
| ReinboT | 59.0 | 40.4 | 56.6 | 38.8 | 48.7 |
| RIPT-VLA | 86.0 | 72.2 | 82.2 | 72.2 | 78.2 |
| **RobustVLA** | 88.2 | 68.2 | **83.0** | 75.2 | 78.7 |
| **RobustVLA-C** | **89.6** | **82.0** | 79.2 | **77.6** | **82.1** |

**Observation Perturbation Settings.** To evaluate the model's ability to cope with uncertainty in environmental observations, we perturb both the first- and third-view inferences during autonomous interaction. The average success rates are shown in Tab. 1 and Appendix Tab. 4. The results show that our proposed method has the best resistance to observation perturbations compared to all other baselines, with average SR of 82.5% and 82.2% for RobustVLA and RobustVLA-C. Among the Offline IL models, OpenVLA-OFT (80.6%) shows a significant improvement over the original Open-VLA (47.9%). Offline RL algorithms, however, perform less well, with similar success rates (all around 60%). This highlights the importance of the VLA model's autonomous interaction with the environment and the Jacobian regularization term in robustly resisting observation perturbations.

**Action Perturbation Setting.** We add zero-mean Gaussian noise to the inferred actions during the interaction between the VLA model and the environment to test the model's resilience to action noise. The average success rates for the four LIBERO suites are shown in Tab. 2 and Appendix Tab. 5. The results demonstrate that even under varying degrees of action noise, our method achieves the best performance compared to all baselines. The proposed RobustVLA and RobustVLA-C achieved similar performance, at 54.8% and 54.7%, respectively. OpenVLA-OPT (53.5%) performed best in the offline IL category and outperformed the best Offline RL method, ARFM (50.1%). This demonstrates that our proposed method improves the VLA model's resilience to action perturbations and robust decision execution.

**Joint Perturbation Settings.** To thoroughly evaluate the VLA model's robustness against environmental perturbations, we further set a more challenging joint perturbation: observation image rotation and an action noise level of 0.1. The average success rates for the four LIBERO suites are shown in Tab. 3. The results show that the proposed RobustVLA-C using curriculum learning (82.1%) achieves the best performance, significantly outperforming other methods. Furthermore, online RL algorithms generally outperform both offline IL and offline RL algorithms. Among offline IL algorithms, OpenVLA-OFT (69.5%) performs best, while OpenVLA (only 10.8%) performs worst. Meanwhile, ReinboT (48.7%) slightly outperforms other offline RL algorithms. These experimental results highlight several important conclusions: **1)** autonomous online RL interaction

can enhance model robustness in the face of complex environmental perturbations; **2)** our proposed Jacobian penalty and action smoothness constraint can effectively guide the VLA model to cope with environmental uncertainty; and **3)** in challenging OOD scenarios, using curriculum learning allows the VLA model to more gradually adapt to downstream tasks and cope with environmental perturbations, thereby generating robust and stable decision-making behavior.

## 5.2 Transfer Learning Performance Comparison

In this section, we examine the transfer learning capabilities of RobustVLA to target domains containing environmental uncertainty. To this end, we transfer the pre-trained VLA model on the LIBERO Goal suite to downstream tasks with environmental perturbations (image rotation and $0.15$ action noise level). The results in Fig. 3 show that the performance of the baseline RIPT-VLA decreases with increasing number of environment rollouts, falling short of the proposed RobustVLA. This suggests that in perturbed scenarios, online RL fine-tuning, which simply maximizes the ultimate success reward, is insufficient to cope with environmental uncertainty and perturbations.

In contrast, on the *open drawer* and *put bowl* tasks, RobustVLA achieves improvements of $8.0\%$ and $16.0\%$ over direct zero-shot transfer, respectively. This demonstrates that the proposed RobustVLA can significantly withstand environmental perturbations and adopt more robust and stable behaviors after only a few environment rollouts, thus exhibiting superior transfer learning capabilities. More results in Appendix Fig. 6.

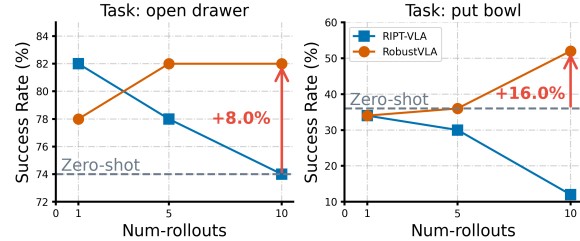

Figure 3: Comparison of transfer learning capabilities in OOD scenarios with uncertainty.

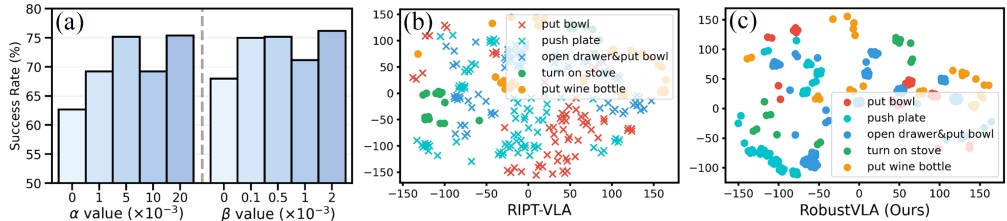

Figure 4: (a) Ablation studies on Jacobian weight $\alpha$, and action-smooth weight $\beta$. (b-c) T-SNE visualization of the observation representations of the baseline RIPT-VLA and the proposed RobustVLA. "•": task success; "×": task failure.

## 5.3 Ablation Analysis

**Hyperparameter Ablation.** In the RobustVLA optimization objective, the hyperparameters $\alpha$ and $\beta$ control the strength of the robustness component. We perform ablation analysis of these two parameters in a joint perturbation setting: LIBERO goal suite, observation image rotation, and an action noise level of $0.1$. Results in Fig. 4(a) show that the omission of either penalty term leads to a drop in VLA model performance, while our algorithm exhibits good hyperparameter insensitivity.

**Observation Representation Visualization.** To investigate how the proposed penalty objective affects the observation representation of the VLA model, we performed a T-SNE (Maaten & Hinton, 2008) visualization analysis in the same joint perturbation setting, as shown in Fig. 4(b-c). The result shows that observation representations corresponding to the RobustVLA remain largely unchanged despite varying degrees of environmental perturbation. This demonstrates its robustness to environmental uncertainty, enabling robust task performance. In contrast, the baseline RIPT-VLA exhibits overly separated clusters in the observation representations of some downstream tasks. This indicates a deviation from the desired trajectory for task performance, leading to a lack of robustness and reliability in decision-making.

**Action Distribution Visualization.** We visualized the action distribution to explore how various noises affect model execution and to compare the proposed RobustVLA with the baseline RIPT-

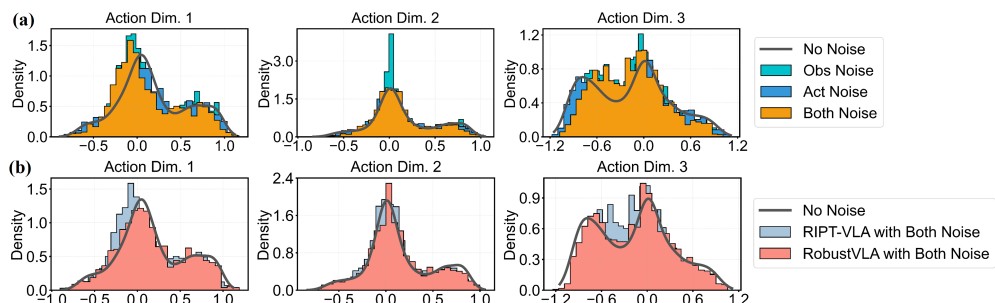

Figure 5: (a) Action distribution during model inference under four noise conditions. (b) Comparison of action distribution between baseline RIPT-VLA and the proposed RobustVLA.

VLA. Specifically, we recorded 100 action trajectories during RIPT-VLA inference in LIBERO Spatial. Four settings were utilized: **1)** no observation noise or action noise; **2)** only image rotation perturbation; **3)** only action noise (noise level 0.1); and **4)** both image rotation perturbation and action noise. The visualization results are shown in Fig. 5(a) and Appendix Fig. 7. The results show that the action trajectories exhibit diverse distribution patterns and coverage under different noise conditions. This indicates that observation perturbation and action perturbation have different emphases and degrees of influence on robot manipulation behavior, especially when both perturbations are present. Furthermore, the action distribution comparison between the baseline RIPT-VLA and the proposed RobustVLA is shown in Fig. 5(b) and Appendix Fig. 8. The results show that under joint perturbation, the action distribution of RobustVLA is closer to the action distribution under noise-free conditions compared to RIPT-VLA. This demonstrates that RobustVLA can adaptively adjust its behavior, enabling the robot to better resist environmental disturbances and thus produce more robust and stable decision-making actions.

## 6 CONCLUSION

In this work, we investigate how VLA models can effectively cope with environmental uncertainty. While large-scale pre-training endows VLA models with flexible manipulation skills, environmental noise can render typical online post-training unreliable when applied to downstream tasks. We therefore propose RobustVLA, a robustness-aware online RL post-training method for VLA models. We establish robustness bounds and induce a concise robust optimization objective. Extensive experiments confirm that our method significantly improves the stability and robustness of VLA models. These findings highlight that robustness-aware post-training is a key step towards reliable model deployment. A promising future work is to investigate the robustness of VLA models to other types of perturbations during autonomous interaction, such as dynamic shift and external disturbances.

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

# A    APPENDIX

## A.1    MORE RELATED WORK

**Vision-Language-Action Models.**    Recent advances in foundation models have extended large-scale pretraining beyond language and vision to embodied agents, giving rise to Vision-Language-Action (VLA) models. The initial wave of this paradigm was marked by two key approaches: extending robotics-native architectures to internet scale, exemplified by RT-1 (Brohan et al., 2022) and RT-2 (Zitkovich et al., 2023), and injecting multimodal sensor data directly into large language models, with PaLM-E (Driess et al., 2023) representing a landmark embodied language model. Subsequent efforts explored improved architectures and scaling laws, with models such as Octo (Team et al., 2024) and OpenVLA (Kim et al., 2024) establishing powerful, open-source benchmarks. Architectural exploration also led to new policy representations, exemplified by the use of diffusion transformers in Dita (Hou et al., 2025) for scaling generalist policies. More recent work has intensified the focus on grounding VLAs in high-quality, robotics-specific data, including the aggregation of diverse datasets via projects like RT-X (O'Neill et al., 2024), GR-3 (Cheang et al., 2025), and pioneering data collection methods such as Mobile ALOHA (Fu et al., 2024). Meanwhile, to advance real-world generalization, models including H-RDT (Bi et al., 2025), $\pi_0$ (Black et al., 2024) and $\pi_{0.5}$ (Intelligence et al., 2025) have been developed to improve embodied reasoning and address open-world, knowledge-driven tasks. Collectively, these studies position VLA models as a core foundation for general-purpose embodied intelligence. Moreover, in the field of robust vision-language manipulation, previous work, GEVRM (Zhang et al., 2025a), evaluates environmental perturbations by simulating responses and optimizing them through prototype contrastive learning. However, GEVRM is inherently a hierarchical imitation learning framework, and the distribution shift of its visual planning objectives during training and testing can lead to performance bottlenecks. Other work (Hancock et al., 2025) has proposed a runtime intervention scheme that identifies model-sensitive input regions and modifies task-irrelevant regions to reduce model sensitivity, thereby improving visual robustness. However, this approach relies heavily on off-the-shelf VLMs and segmentation models, and its primary limitation is accurately distinguishing objects from background clutter. In contrast to these works, our work focuses on online RL post-training to further adapt and coordinate the VLA model with downstream tasks.

## A.2    FURTHER ANALYSIS OF MODEL ROBUSTNESS

Building upon the upper bounds established in the theorems, we next examine two corollaries that sharpen the theoretical insights under additional structural assumptions. These corollaries illustrate how the abstract bounds specialize in contractive dynamics or strongly regularized update regimes, thereby revealing concrete conditions under which robustness can be guaranteed at scale.

**Corollary 1.** *Under the assumptions of Theorem 1 but with $\epsilon_{\text{offline}} = 0$ (the imitation model is perfect) and contractive dynamics $L_f < 1$, the expected return gap satisfies*

$$\mathbb{E}\big[J(\pi^*) - J(\pi)\big] \;\leq\; H \cdot L_r \cdot \frac{1}{1 - L_f} \cdot \lambda\epsilon_s.$$

Corollary 1 shows that the compounding factor $\mathcal{O}(L_f^H)$ collapses to a linear dependence on the rollout horizon $H$, yielding a return gap on the order of $H \cdot \lambda\epsilon_s$. What this implies is that observation noise no longer causes exponential amplification. This highlights a regime where Jacobian regularization has direct leverage $--$ by shrinking $\lambda$, the sensitivity to $\epsilon_s$ is proportionally reduced, ensuring that perception errors do not destabilize the system even in long rollouts. In practice, this means that in sufficiently stable environments, robustness to noisy observations is achievable with only modest regularization strength.

**Corollary 2.** *Under the setting of Theorem 3 but with discounted return $J(\pi) = \mathbb{E}\big[\sum_{t \geq 1} \gamma^{t-1} r(s_t, a_t)\big]$, for some discount factor $\gamma \in (0, 1)$ with contractive dynamics $L_f \in (0, 1)$. Assume the per-step driving term $\zeta_t := \lambda\epsilon_s + \epsilon_{\text{offline}} + \sum_{i=1}^N \delta_i + \|\xi_t\|$ is uniformly bounded in expectation, i.e. there exists $C < \infty$ such that $\sup_t \mathbb{E}[\zeta_t] \leq C$, and that $\sup_t \mathbb{E}[\epsilon_t] \leq E < \infty$. Then the discounted expected return gap is bounded independently of the rollout horizon $H$:*

$$\mathbb{E}\big[J(\pi^*) - J(\pi)\big] = \mathcal{O}(1).$$

Corollary 2 establishes that in the discounted setting with contractive dynamics ($L_f < 1$), the expected return gap remains bounded by a constant that does not depend on the rollout horizon $H$. Intuitively, discounting attenuates the influence of long-term deviations, while contractive dynamics prevent the state deviation $d_t$ from growing unboundedly even when both observation and action perturbations are present. A key observation is that the nonstationarity term $\sum_{i=1}^{N} \delta_i$ enters the bound only through the uniform driving term $\zeta_t$; as long as these accumulated model updates remain bounded, the discounted error cannot compound over time. Consequently, the effect of both perturbations and update-induced drift is geometrically damped by the factor $\gamma L_f < 1$, guaranteeing that the overall return difference remains $\mathcal{O}(1)$ even as $H \to \infty$. Practically, this indicates that stable long-horizon adaptation is achievable provided that policy updates are sufficiently smooth and that the system dynamics do not amplify perturbations.

Together, these two corollaries demonstrate how regularization directly modulates the scaling of the return gap: Jacobian regularization turns exponential amplification into linear growth under stable dynamics, while smoothness regularization caps compounding drift at a constant level under controlled updates. This dual perspective makes clear that robustness in VLA fine-tuning is not an abstract desideratum but a quantitatively attainable property under well-chosen regularization regimes.

## A.3 PROOF OF ROBUSTNESS THEORIES

**Proof of Theorem 1 (Error Amplification Bound)**

**Lemma 1** (State Deviation Recursion). *Let $d_t := \|s_t - s_t^*\|$ denote the state deviation at time $t$ between $\pi$ and $\pi^*$. Then under Lipschitz dynamics $f$, we have:*

$$d_{t+1} \leq L_f d_t + L_f(\lambda \epsilon_s + \epsilon_{\text{offline}}).$$

***Proof.*** At time $t$, the model $\pi$ acts on perturbed state $\tilde{s}_t$:

$$a_t = \pi(\tilde{s}_t), \quad a_t^* = \pi^*(s_t).$$

We decompose:

$$\begin{aligned}
\|a_t - a_t^*\| &\leq \|\pi(\tilde{s}_t) - \pi(s_t)\| + \|\pi(s_t) - \pi^*(s_t)\| \\
&\leq \lambda \cdot \|\tilde{s}_t - s_t\| + \|\pi - \pi^*\|_\infty \\
&\leq \lambda \epsilon_s + \epsilon_{\text{offline}}.
\end{aligned}$$

Using Lipschitz continuity of $f$, the dynamics step gives:

$$d_{t+1} = \|f(s_t, a_t) - f(s_t^*, a_t^*)\| \leq L_f \cdot (\|s_t - s_t^*\| + \|a_t - a_t^*\|),$$

which gives the recurrence:

$$d_{t+1} \leq L_f d_t + L_f(\lambda \epsilon_s + \epsilon_{\text{offline}}).$$

$\square$

**Theorem A.1** (Restatement of Theorem 1). *Assume perturbed observations $\tilde{s}_t = s_t + \delta_s^t$ with $\|\delta_s^t\| \leq \epsilon_s$, and bounded Jacobian $\|\nabla_s \pi_t(s)\| \leq \lambda$. Then:*

$$\mathbb{E}\big[J(\pi^*) - J(\pi)\big] \leq \mathcal{O}(H L_r L_f^H) \cdot (\epsilon_{\text{offline}} + \lambda \epsilon_s).$$

***Proof.*** From Lemma 1, the state deviation satisfies, for $d_0 = 0$,

$$d_t \leq (\lambda \epsilon_s + \epsilon_{\text{offline}}) L_f \sum_{j=0}^{t-1} L_f^j = (\lambda \epsilon_s + \epsilon_{\text{offline}}) L_f \frac{L_f^t - 1}{L_f - 1} = \mathcal{O}(L_f^t)(\lambda \epsilon_s + \epsilon_{\text{offline}}).$$

At each step, the reward difference obeys the Lipschitz bound

$$|r(s_t, a_t) - r(s_t^*, a_t^*)| \leq L_r(d_t + \lambda \epsilon_s + \epsilon_{\text{offline}}).$$

Taking expectations on both sides and using the deterministic upper bound on $d_t$, we obtain

$$\mathbb{E}\big[|r(s_t, a_t) - r(s_t^*, a_t^*)|\big] \leq L_r(d_t + \lambda \epsilon_s + \epsilon_{\text{offline}}).$$

Summing over $t = 1, \ldots, H$ gives

$$\mathbb{E}\big[J(\pi^*) - J(\pi)\big] \ \leq \ \sum_{t=1}^{H} L_r\big(d_t + \lambda\epsilon_s + \epsilon_{\text{offline}}\big).$$

Finally substituting $d_t = \mathcal{O}(L_f^t)(\lambda\epsilon_s + \epsilon_{\text{offline}})$ and evaluating the geometric series yields

$$\mathbb{E}\big[J(\pi^*) - J(\pi)\big] \ \leq \ \mathcal{O}(HL_rL_f^H)\big(\lambda\epsilon_s + \epsilon_{\text{offline}}\big),$$

which completes the proof. $\qquad\square$

**Proof of Theorem 2 (Return Drift Control)**

**Lemma 2** (State Deviation under Smooth Regularization + Action Noise)**.** *Let* $d_t = \|s_t - s_t^*\|$. *Under* $L_f$-*Lipschitz dynamics and executed action* $a_t = \pi_t(s_t) + \xi_t$,

$$d_{t+1} \leq L_f\big(d_t + \epsilon_t + \|\xi_t\|\big),$$

*with* $\epsilon_t \leq \epsilon_{\text{offline}} + \sum_{i=1}^{N} \delta_i$.

*Proof.* We decompose the action error:

$$\|a_t - a_t^*\| = \|\pi_t(s_t) + \xi_t - \pi^*(s_t)\| \leq \|\pi_t(s_t) - \pi^*(s_t)\| + \|\xi_t\| = \epsilon_t + \|\xi_t\|.$$

Then the Lipschitz dynamics give:

$$d_{t+1} = \|f(s_t, a_t) - f(s_t^*, a_t^*)\| \leq L_f(\|s_t - s_t^*\| + \|a_t - a_t^*\|) \leq L_f(d_t + \epsilon_t + \|\xi_t\|).$$

$\qquad\square$

**Lemma 3** (Expected Norm of Gaussian Noise)**.** *Let* $\xi_t \sim \mathcal{N}(0, \sigma^2 I_d)$ *be a* $d$-*dimensional isotropic Gaussian random vector. Then the expected Euclidean norm satisfies:*

$$\mathbb{E}[\|\xi_t\|] \leq \sigma \cdot \sqrt{d}.$$

*Proof.* Let $z_i := \xi_{t,i}/\sigma \sim \mathcal{N}(0,1)$, so $\xi_{t,i} = \sigma z_i$. Then the norm becomes:

$$\|\xi_t\| = \left(\sum_{i=1}^{d} \sigma^2 z_i^2\right)^{1/2} = \sigma \cdot \left(\sum_{i=1}^{d} z_i^2\right)^{1/2}.$$

Let $Z := \sum_{i=1}^{d} z_i^2$, then $Z \sim \chi^2(d)$ is a chi-squared distribution with $d$ degrees of freedom. So the norm becomes:

$$\|\xi_t\| = \sigma \cdot \sqrt{Z} \quad \Rightarrow \quad \mathbb{E}[\|\xi_t\|] = \sigma \cdot \mathbb{E}[\sqrt{Z}].$$

Although $\mathbb{E}[\sqrt{Z}]$ has no closed form, we apply Jensen's inequality. Since the square root function is concave and $Z \geq 0$,

$$\mathbb{E}[\sqrt{Z}] \leq \sqrt{\mathbb{E}[Z]} = \sqrt{d}.$$

Hence,

$$\mathbb{E}[\|\xi_t\|] \leq \sigma \cdot \sqrt{d}.$$

$\qquad\square$

**Theorem A.2** (Restatement of Theorem 2)**.** *Assume actions are perturbed as* $a_t = \pi_t(s_t) + \xi_t$ *with* $\xi_t \sim \mathcal{N}(0, \sigma^2 I_d)$, *and VLA models satisfy* $\|\pi_i - \pi_{i-1}\|_\infty \leq \delta_i$. *Then:*

$$\mathbb{E}\big[J(\pi^*) - J(\pi)\big] \leq \mathcal{O}(HL_rL_f^H) \cdot \Big(\epsilon_{\text{offline}} + \sum_{i=1}^{N} \delta_i + \sigma\sqrt{d}\Big).$$

*Proof.* Start from the per-step recurrence (Lemma 2):

$$d_{t+1} \leq L_f\big(d_t + \epsilon_t + \|\xi_t\|\big),$$

and recall the uniform bound on model mismatch $\epsilon_t \leq \epsilon_{\text{offline}} + \sum_{i=1}^{N} \delta_i$.

Unrolling the linear recurrence with $d_0 = 0$ gives

$$d_t \leq \sum_{j=0}^{t-1} L_f^{t-j}(\epsilon_j + \|\xi_j\|) \leq \sum_{j=0}^{t-1} L_f^{t-j}\Big(\epsilon_{\text{offline}} + \sum_{i=1}^{N} \delta_i + \|\xi_j\|\Big).$$

Take expectation on both sides and use linearity:

$$\mathbb{E}[d_t] \leq \sum_{j=0}^{t-1} L_f^{t-j}\Big(\epsilon_{\text{offline}} + \sum_{i=1}^{N} \delta_i + \mathbb{E}\|\xi_j\|\Big).$$

Now apply Lemma 3, which yields $\mathbb{E}\|\xi_j\| \leq \sigma\sqrt{d}$ for every $j$. Consequently,

$$\mathbb{E}[d_t] \leq \Big(\epsilon_{\text{offline}} + \sum_{i=1}^{N} \delta_i + \sigma\sqrt{d}\Big) \sum_{j=0}^{t-1} L_f^{t-j} = \Big(\epsilon_{\text{offline}} + \sum_{i=1}^{N} \delta_i + \sigma\sqrt{d}\Big) \cdot \mathcal{O}(L_f^t),$$

where the finite geometric sum produces a factor of order $\mathcal{O}(L_f^t)$.

Because the reward is $L_r$-Lipschitz,

$$|r(s_t, a_t) - r(s_t^*, a_t^*)| \leq L_r\big(d_t + \epsilon_t + \|\xi_t\|\big).$$

Taking expectation and summing over $t = 0, \ldots, H - 1$,

$$\mathbb{E}\big[J(\pi^*) - J(\pi)\big] \leq L_r \sum_{t=0}^{H-1} \mathbb{E}[d_t] + L_r \sum_{t=0}^{H-1} \Big(\epsilon_{\text{offline}} + \sum_{i=1}^{N} \delta_i + \mathbb{E}\|\xi_t\|\Big)$$

$$\leq L_r \sum_{t=0}^{H-1} \mathcal{O}(L_f^t)\Big(\epsilon_{\text{offline}} + \sum_{i=1}^{N} \delta_i + \sigma\sqrt{d}\Big) + L_r H\Big(\epsilon_{\text{offline}} + \sum_{i=1}^{N} \delta_i + \sigma\sqrt{d}\Big).$$

Collecting geometric-series terms gives the claimed form

$$\mathbb{E}\big[J(\pi^*) - J(\pi)\big] \leq \mathcal{O}(HL_r L_f^H) \cdot \Big(\epsilon_{\text{offline}} + \sum_{i=1}^{N} \delta_i + \sigma\sqrt{d}\Big).$$

$\square$

### Proof of Theorem 3 (Robust Stability Guarantee)

**Lemma 4** (State Deviation with Nested Perturbation). *Let $d_t = \|s_t - s_t^*\|$. Under $L_f$-Lipschitz dynamics, and with both observation perturbations $\|\tilde{s}_t - s_t\| \leq \epsilon_s$ and action noise $\xi_t$, the state deviation satisfies*

$$d_{t+1} \leq L_f \cdot d_t + L_f\big(\lambda\epsilon_s + \epsilon_t + \|\xi_t\|\big),$$

*where, for all $t$, $\epsilon_t := \|\pi_t - \pi^*\| \leq \epsilon_{\text{offline}} + \sum_{i=1}^{N} \delta_i$.*

*Proof.* Decompose the action error at time $t$:

$$\|a_t - a_t^*\| = \|\pi_t(\tilde{s}_t) + \xi_t - \pi^*(s_t)\|$$
$$\leq \|\pi_t(\tilde{s}_t) - \pi_t(s_t)\| + \|\pi_t(s_t) - \pi^*(s_t)\| + \|\xi_t\|$$
$$\leq \lambda\|\tilde{s}_t - s_t\| + \epsilon_t + \|\xi_t\| \leq \lambda\epsilon_s + \epsilon_t + \|\xi_t\|.$$

Applying $L_f$-Lipschitz dynamics,

$$d_{t+1} = \|f(s_t, a_t) - f(s_t^*, a_t^*)\| \leq L_f\big(\|s_t - s_t^*\| + \|a_t - a_t^*\|\big),$$

which yields the stated recursion. $\square$

**Theorem A.3** (Restatement of Theorem 3). *Under both observation and action perturbations, and both Jacobian and smooth regularizations, the return gap satisfies:*

$$\mathbb{E}[J(\pi^*) - J(\pi)] \leq \mathcal{O}(HL_rL_f^H)\Big(\epsilon_{\text{offline}} + \sum_{i=1}^{N}\delta_i + \lambda\epsilon_s + \sigma\sqrt{d}\Big).$$

*Proof.* From Lemma 4, the state deviation satisfies

$$d_{t+1} \leq L_fd_t + L_f\big(\lambda\epsilon_s + \epsilon_t + \|\xi_t\|\big), \quad \epsilon_t \leq \epsilon_{\text{offline}} + \sum_{i=1}^{N}\delta_i.$$

Unrolling the linear recurrence with $d_0 = 0$ yields

$$d_t \leq \sum_{j=0}^{t-1}L_f^{t-1-j}\Big(\lambda\epsilon_s + \epsilon_{\text{offline}} + \sum_{i=1}^{N}\delta_i + \|\xi_j\|\Big).$$

Taking expectations and using linearity,

$$\mathbb{E}[d_t] \leq \sum_{j=0}^{t-1}L_f^{t-1-j}\Big(\lambda\epsilon_s + \epsilon_{\text{offline}} + \sum_{i=1}^{N}\delta_i + \mathbb{E}\|\xi_j\|\Big).$$

Applying Lemma 3, $\mathbb{E}\|\xi_j\| \leq \sigma\sqrt{d}$, gives

$$\mathbb{E}[d_t] \leq \Big(\lambda\epsilon_s + \epsilon_{\text{offline}} + \sum_{i=1}^{N}\delta_i + \sigma\sqrt{d}\Big)\sum_{j=0}^{t-1}L_f^{t-1-j}.$$

Thus,

$$\mathbb{E}[d_t] \leq \mathcal{O}(L_f^t)\Big(\lambda\epsilon_s + \epsilon_{\text{offline}} + \sum_{i=1}^{N}\delta_i + \sigma\sqrt{d}\Big).$$

Because rewards are $L_r$-Lipschitz,

$$|r(s_t, a_t) - r(s_t^*, a_t^*)| \leq L_r\big(d_t + \lambda\epsilon_s + \epsilon_t + \|\xi_t\|\big).$$

Similar to the proof of Theorem 2, we obtain:

$$\mathbb{E}[J(\pi^*) - J(\pi)] \leq \mathcal{O}(HL_rL_f^H)\Big(\epsilon_{\text{offline}} + \sum_{i=1}^{N}\delta_i + \lambda\epsilon_s + \sigma\sqrt{d}\Big).$$

$\square$

**Proof of Corollaries 1 and 2**

**Corollary 3** (Restatement of Corollary 1). *Under the assumptions of Theorem 1 but with $\epsilon_{\text{offline}} = 0$ (the imitation model is perfect) and contractive dynamics $L_f < 1$, the expected return gap satisfies*

$$\mathbb{E}\big[J(\pi^*) - J(\pi)\big] \leq H \cdot L_r \cdot \frac{1}{1 - L_f} \cdot \lambda\epsilon_s.$$

*Proof.* With $\epsilon_{\text{offline}} = 0$, Lemma 1 gives (for $d_0 = 0$)

$$d_t \leq L_f(\lambda\epsilon_s)\sum_{i=0}^{t-1}L_f^i = \lambda\epsilon_s \cdot L_f \cdot \frac{1 - L_f^t}{1 - L_f}.$$

Since $L_f \in (0, 1)$, the factor $L_f\frac{1-L_f^t}{1-L_f} \leq \frac{L_f}{1-L_f}$ uniformly in $t$. Thus

$$d_t \leq \lambda\epsilon_s \cdot \frac{L_f}{1 - L_f}.$$

The per-step reward difference satisfies

$$|r(s_t, a_t) - r(s_t^*, a_t^*)| \leq L_r\big(d_t + \lambda\epsilon_s\big) \leq L_r\lambda\epsilon_s\left(\frac{L_f}{1 - L_f} + 1\right) = L_r\lambda\epsilon_s \cdot \frac{1}{1 - L_f}.$$

Taking expectations and summing over $t = 1, \ldots, H$ yields

$$\mathbb{E}\big[J(\pi^*) - J(\pi)\big] \leq H \cdot L_r\lambda\epsilon_s \cdot \frac{1}{1 - L_f},$$

which is the stated bound. □

**Corollary 4** (Restatement of Corollary 2). *Under the setting of Theorem 3 but with discounted return $J(\pi) = \mathbb{E}\Big[\sum_{t \geq 1} \gamma^{t-1}r(s_t, a_t)\Big]$, for some discount factor $\gamma \in (0, 1)$ with contractive dynamics $L_f \in (0, 1)$. Assume the per-step driving term $\zeta_t := \lambda\epsilon_s + \epsilon_{\text{offline}} + \sum_{i=1}^N \delta_i + \|\xi_t\|$ is uniformly bounded in expectation, i.e. there exists $C < \infty$ such that $\sup_t \mathbb{E}[\zeta_t] \leq C$, and that $\sup_t \mathbb{E}[\epsilon_t] \leq E < \infty$. Then the discounted expected return gap is bounded independently of the rollout horizon $H$:*

$$\mathbb{E}\big[J(\pi^*) - J(\pi)\big] = \mathcal{O}(1).$$

*Proof.* From Lemma 4 we have the one-step recursion

$$d_{t+1} \leq L_f d_t + L_f\, \zeta_t,$$

where $d_t = \|s_t - s_t^*\|$. Unrolling with $d_0 = 0$ gives

$$d_t \leq \sum_{j=0}^{t-1} L_f^{t-1-j}\, L_f\, \zeta_j = \sum_{j=0}^{t-1} L_f^{t-j}\, \zeta_j.$$

Taking expectations and using $\mathbb{E}[\zeta_j] \leq C$ for all $j$ yields

$$\mathbb{E}[d_t] \leq C\sum_{j=0}^{t-1} L_f^{t-j} = C\sum_{k=1}^t L_f^k = C \cdot \frac{L_f(1 - L_f^t)}{1 - L_f} \leq C \cdot \frac{L_f}{1 - L_f},$$

so $\mathbb{E}[d_t]$ is uniformly bounded in $t$.

The per-step expected reward difference satisfies (by $L_r$-Lipschitzness of $r$ and the definition of $\zeta_t$)

$$\mathbb{E}\big[|r(s_t, a_t) - r(s_t^*, a_t^*)|\big] \leq L_r\big(\mathbb{E}[d_t] + \mathbb{E}[\zeta_t]\big) \leq L_r\left(C \cdot \frac{L_f}{1 - L_f} + C\right) = L_r C\left(\frac{L_f}{1 - L_f} + 1\right).$$

Therefore

$$\mathbb{E}[J(\pi^*) - J(\pi)] \leq L_r\sum_{t \geq 1} \gamma^{t-1}\Big(\mathbb{E}[d_t] + \mathbb{E}[\zeta_t]\Big)$$

$$\leq L_r C\sum_{t \geq 1} \gamma^{t-1}\left(\frac{L_f(1 - L_f^t)}{1 - L_f} + 1\right).$$

Evaluate the geometric series (using $\gamma L_f < 1$):

$$\sum_{t \geq 1} \gamma^{t-1}\frac{L_f(1 - L_f^t)}{1 - L_f} = \frac{L_f}{1 - L_f}\sum_{t \geq 1}\big(\gamma^{t-1} - (\gamma L_f)^{t-1}\big) = \frac{L_f}{1 - L_f}\left(\frac{1}{1 - \gamma} - \frac{1}{1 - \gamma L_f}\right),$$

and

$$\sum_{t \geq 1} \gamma^{t-1} = \frac{1}{1 - \gamma}.$$

Therefore

$$\mathbb{E}[J(\pi^*) - J(\pi)] \leq L_r C\left(\frac{L_f}{1 - L_f}\left(\frac{1}{1 - \gamma} - \frac{1}{1 - \gamma L_f}\right) + \frac{1}{1 - \gamma}\right),$$

which is a finite constant independent of the rollout horizon $H$. Thus, we obtain:

$$\mathbb{E}[J(\pi^*) - J(\pi)] = \mathcal{O}(1).$$

□

## A.4 Observation perturbation setting

Following previous work, GEVRM (Zhang et al., 2025a), we provide details on implementing observation perturbations in the LIBERO simulation platform: **1) Image Shift:** The image state is randomly shifted to the upper left, with a maximum translation rate of 0.3 relative to the image size. **2) Image Rotation:** The image state is randomly rotated counterclockwise with a maximum rotation angle of 30 degrees. **3) Color Jitter:** The image state saturation, brightness, contrast, and sharpness are randomly jittered with a maximum random factor of 3. **4) Image Occlusions:** The image state is randomly occluded with a random number of occlusion blocks ranging from 1 to 3 and a maximum length of 20. **5) Image Erasing:** The image state is perturbed by random noise blocks with a maximum size of 0.1 of the original image. We add observation perturbations to both the first-view and third-view observation images, which makes the constructed robust benchmark platform more challenging.

## A.5 Implementation Details

**Reward Densification.** Given that only sparse 0-1 rewards representing task success or failure are directly obtained from the environment, we follow the previous work, ReinboT (Zhang et al., 2025c), to densify the rewards. Specifically, in addition to the task completion reward obtained from the environment, we consider ten additional rewards, as shown in Tab. 7. The ORB-related reward from ReinboT is not considered, as its computation is somewhat time-consuming during online interaction.

**Experimental implementation.** We conduct our experiments on OpenVLA-OFT (Kim et al., 2025) using the official checkpoints as the base model, with RIPT-VLA's LoRA adaptor checkpoints (Tan et al., 2025b) applied for adaptation. The implementation is based on the RIPT-VLA codebase (Tan et al., 2025b). Moreover, $\mathcal{R}_{\text{Jac}}$ requires computing the two-norm of $\nabla_s \log \pi_\theta(a|s)$, which is expensive and memory-intensive to compute directly at the image pixels. Thus, we compute the Jacobian on the low-dimensional embeddings obtained from the Llama-2 encoding used by OpenVLA-OFT. Training is performed on a single GPU with LoRA of rank 32. The post-training procedure takes approximately 24 hours for fine-tuning. The training parameters are shown in Tab. 6, and other parameters follow the previous work, RIPT-VLA (Tan et al., 2025b). All experiments are conducted on the following hardware: **CPU:** Intel(R) Xeon(R) Platinum 8358 @ 2.60GHz; **GPU:** NVIDIA A100-SXM4-80GB.

**Baseline algorithm reproduction.** To replicate the baseline algorithms, $\pi_0$, ReinboT, RWR, and ARFM are all implemented in the repositories: *huggingface/lerobot* (Cadene et al., 2024). To replicate GEVRM, we utilize the same prototype contrastive learning method for observation representation in the RIPT-VLA code. For other baseline models, we utilize the official implementation.

## A.6 Use of Large Language Models

During the preparation of this manuscript, we employed OpenAI's ChatGPT (GPT-5) to assist with writing refinement. The model was used exclusively for polishing the language, improving clarity, and suggesting minor stylistic alternatives. All technical content, including problem formulation, theoretical analysis, algorithm design, and experimental results, was conceived, derived, and validated by the authors. The use of ChatGPT did not influence the scientific claims, results, or conclusions presented in this work.

Table 4: Detailed comparison of the average SR (%) of the four LIBERO suites under observation perturbations. The best result is highlighted in bold, and the second-best result is underlined. Here, "OpenVLA*" denotes the OpenVLA-OFT model, "Ours" refers to RobustVLA model, and "Ours-C" refers to RobustVLA-C model.

| Pertur-bation | Tasks | $\pi_0$ | ReinboT | RWR | ARFM | RIPT-VLA | OpenVLA* | OpenVLA | GEVRM | Ours | Ours-C |
|---|---|---|---|---|---|---|---|---|---|---|---|
| Shift | Spatial | 35.2 | 21.4 | 18.6 | 11.0 | 51.6 | 54.0 | 38.4 | **54.3** | 54.0 | 46.4 |
| | Goal | 32.8 | 23.6 | 25.2 | 20.6 | 17.4 | 19.4 | 20.0 | **45.0** | 21.6 | 24.8 |
| | Object | 23.4 | 6.0 | 4.6 | 3.6 | 69.8 | 71.4 | 30.8 | 46.0 | 72.2 | **73.6** |
| | Long | 33.2 | 28.2 | 16.8 | 12.4 | 19.0 | 15.9 | 12.2 | **40.2** | 20.4 | 19.6 |
| Rotation | Spatial | 61.8 | 59.6 | 59.4 | 56.8 | 94.2 | 92.6 | 54.0 | 88.1 | **98.4** | 96.4 |
| | Goal | 68.6 | 42.2 | 45.2 | 42.6 | 87.2 | 89.0 | 80.0 | 65.4 | 91.4 | **92.0** |
| | Object | 10.4 | 10.0 | 12.0 | 49.4 | 83.6 | 67.0 | 21.0 | 43.2 | **87.4** | 87.2 |
| | Long | **57.0** | 36.4 | 31.6 | 31.0 | 51.0 | 51.6 | 11.2 | 35.0 | 53.2 | 55.6 |
| Color | Spatial | 81.2 | 79.0 | 81.8 | 79.8 | 97.2 | 98.2 | 56.4 | 86.3 | **98.6** | 97.4 |
| | Goal | 82.0 | 82.0 | 80.4 | 82.4 | 93.2 | **94.0** | 48.0 | 61.2 | 92.4 | 93.5 |
| | Object | 92.2 | 78.2 | 78.0 | 79.8 | 97.0 | 97.0 | 52.5 | 47.0 | **97.4** | 96.2 |
| | Long | 58.4 | 58.4 | 58.8 | 59.8 | 93.0 | **94.4** | 37.2 | 31.0 | 93.8 | 94.0 |
| Occlusions | Spatial | 90.2 | 90.8 | 90.0 | 87.6 | **98.4** | 97.9 | 87.2 | 87.3 | **98.4** | 98.2 |
| | Goal | 89.8 | 93.2 | 89.6 | 91.8 | 95.6 | 94.6 | 65.4 | 64.0 | 95.0 | **97.2** |
| | Object | 95.0 | 94.2 | 94.8 | 95.0 | 96.8 | 97.4 | 81.4 | 48.8 | **98.0** | 96.8 |
| | Long | 78.0 | 78.2 | 78.2 | 77.6 | 89.2 | 91.0 | 68.0 | 39.0 | **91.2** | 89.8 |
| Erasing | Spatial | 85.8 | 85.2 | 86.4 | 86.2 | 97.6 | 97.9 | 85.2 | 91.9 | **98.2** | 98.0 |
| | Goal | 88.8 | 88.4 | 86.4 | 86.2 | 96.8 | 96.4 | 62.5 | 66.8 | **98.2** | 97.2 |
| | Object | 92.0 | 89.7 | 89.6 | 92.4 | 96.8 | 97.4 | 78.0 | 50.6 | **97.6** | 96.8 |
| | Long | 74.2 | 76.2 | 71.8 | 71.0 | 91.2 | **94.0** | 68.5 | 44.0 | 93.4 | 93.8 |
| **Avg.** | | 66.5 | 61.0 | 59.5 | 60.9 | 80.8 | 80.5 | 47.9 | 56.8 | **82.6** | 82.2 |

Table 5: Detailed comparison of the average SR(%) of the four LIBERO suites under action perturbations (noise level = 0.1, 0.2, 0.3). The best result is highlighted in bold, and the second-best result is underlined. "OpenVLA*" denotes the OpenVLA-OFT model, "Ours" refers to RobustVLA model, and "Ours-C" refers to RobustVLA-C model.

| Models | Noise Level 0.1 | | | | | Noise Level 0.2 | | | | | Noise Level 0.3 | | | | |
|---|---|---|---|---|---|---|---|---|---|---|---|---|---|---|---|
| | Spatial | Object | Goal | Long | Avg. | Spatial | Object | Goal | Long | Avg. | Spatial | Object | Goal | Long | Avg. |
| OpenVLA | 61.4 | 57.0 | 50.2 | 46.0 | 53.6 | 61.2 | 42.8 | 36.0 | 16.8 | 39.2 | 27.3 | 7.2 | 11.6 | 0.8 | 11.7 |
| RIPT-VLA | 96.2 | 80.2 | 85.8 | 76.6 | 84.7 | 66.0 | 47.0 | 50.6 | 28.6 | 48.1 | 30.4 | 18.2 | 20.0 | 6.6 | 18.8 |
| OpenVLA* | 95.6 | 84.4 | 87.0 | **82.6** | 87.4 | 71.6 | 50.2 | 51.4 | 36.0 | 52.3 | 36.0 | 16.4 | 25.4 | 5.0 | 20.7 |
| $\pi_0$ | 81.2 | 79.6 | 87.6 | 61.6 | 77.5 | 47.0 | 40.8 | 56.6 | 25.6 | 42.5 | 21.4 | 17.2 | 29.0 | 6.0 | 18.4 |
| RWR | 88.2 | 83.8 | 86.0 | 64.2 | 80.6 | 61.0 | 42.4 | 53.0 | 22.6 | 44.8 | 31.8 | 16.6 | 30.2 | 4.6 | 20.8 |
| ReinboT | 86.2 | 86.0 | 85.6 | 63.2 | 80.3 | 60.2 | 46.4 | **59.2** | 22.2 | 47.0 | 31.2 | 16.6 | **30.6** | 6.4 | 21.2 |
| ARFM | 87.0 | 87.0 | 83.4 | 67.0 | 81.1 | 60.2 | 47.0 | 57.4 | 28.0 | 48.2 | 32.4 | 17.2 | 28.8 | 6.0 | 21.1 |
| **Ours** | **98.2** | **87.4** | **91.0** | 77.2 | 88.5 | **73.4** | 50.6 | 53.8 | 34.6 | 53.1 | **36.8** | **22.4** | 24.2 | 7.4 | **22.7** |
| **Ours-C** | 97.0 | 86.6 | 90.2 | 81.6 | **88.9** | 71.8 | **51.4** | 51.8 | **38.0** | **53.3** | 34.6 | 18.0 | 25.4 | **9.0** | 21.8 |

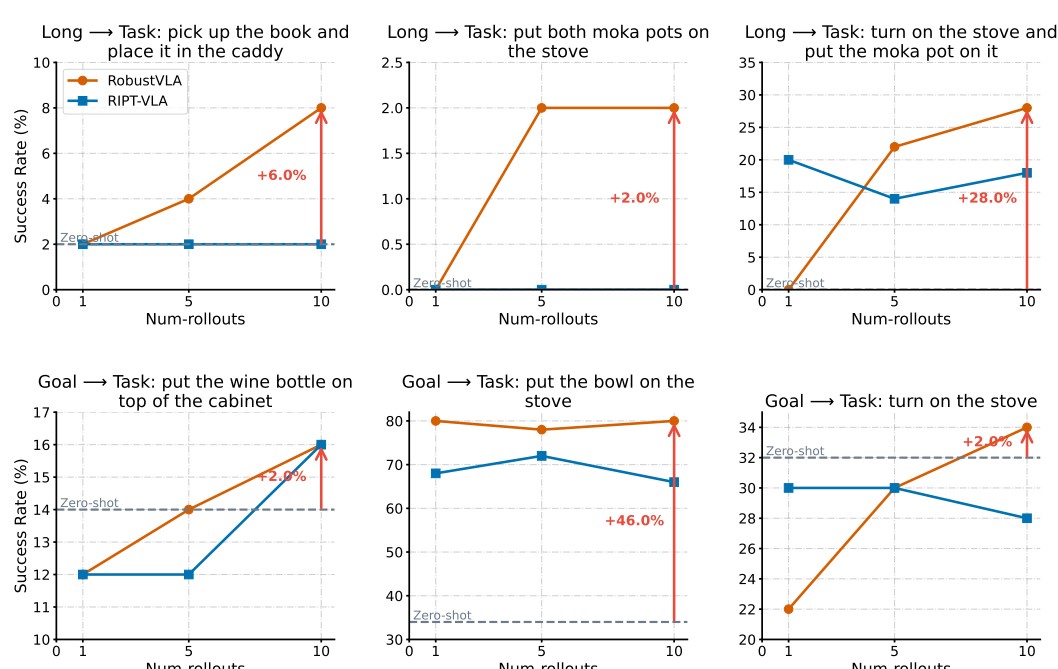

Figure 6: More results on comparison of transfer learning capabilities in OOD tasks with environmental uncertainty (image rotation and $0.15$ action noise level).

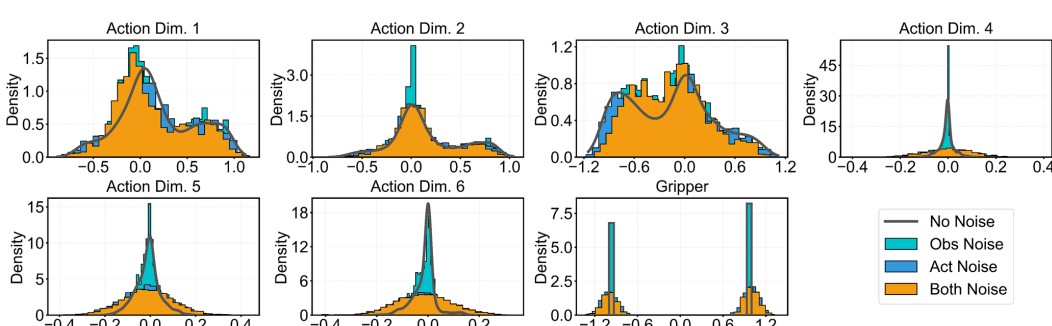

Figure 7: Action distribution during model inference under four noise conditions.

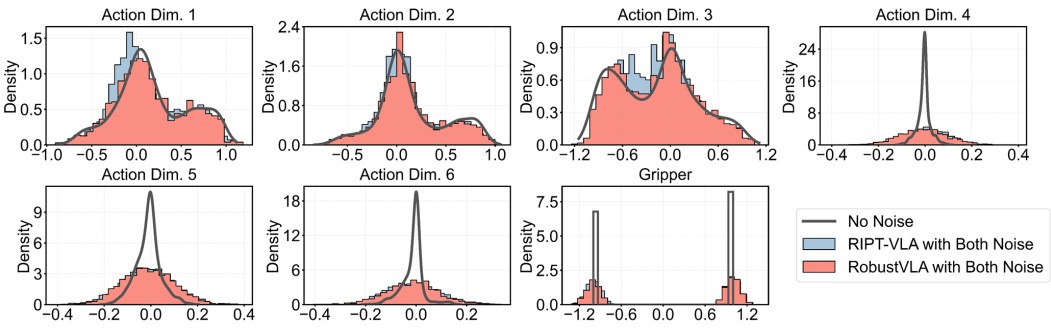

Figure 8: Comparison of action distribution between baseline RIPT-VLA and the proposed RobustVLA.

Table 6: Training hyperparameter configuration.

| Parameter | Value |
|---|---|
| *General* | |
| LoRA Rank | 32 |
| Gradient Accumulation Steps | 1 |
| PPO Epochs | 1 |
| PPO Clip Range | 0.2 |
| PPO Clip High | 0.2 |
| Max Step Batch Size | 2 |
| Learning Rate (LoRA modules) | $1.0 \times 10^{-4}$ |
| Learning Rate (Action head) | $5.0 \times 10^{-5}$ |
| Weight Decay | $1.0 \times 10^{-4}$ |
| Gradient Clip Norm (model) | 1.0 |
| Gradient Clip Norm (header) | 1.0 |
| Total Steps $M$ | 12 |
| Eval Interval $I_{interval}$ | 1 |
| Update Times $N$ | 10 |
| *Robust Regularization Weight* | |
| Jacobian Regularization Weight $\alpha$ | 0.005 |
| Smooth Regularization Weight $\beta$ | 0.0005 |
| *Curriculum Learning* | |
| Success Rate Thresholds ($\tau_{low}, \tau_{high}$) | 0.6, 0.8 |
| Success moving average parameter $\gamma$ | 0.9 |
| Observation Noise Range ($\epsilon_{min,obs}, \epsilon_{max,obs}$) | $[0, 1]$ |
| Action Noise Range ($\epsilon_{min,action}, \epsilon_{max,action}$) | $[0, 0.3]$ |
| Observation Noise Step $\Delta_{obs}$ | 0.2 |
| Action Noise Step $\Delta_{action}$ | 0.02 |
| Probability of no perturbation | 0.15 |

Table 7: Dense reward components and weights.

| Reward Component | Weight |
|---|---|
| *Sub-goal Achievement* | |
| Image MSE ($e^{f_{\text{MSE}}(o_t, o_t^*)}$) | 0.1/10 |
| Image SSIM ($e^{f_{\text{SSIM}}(o_t, o_t^*)}$) | 0.1/10 |
| Gripper Image MSE ($e^{f_{\text{MSE}}(o_t, o_t^*)}$) | 0.1/10 |
| Gripper Image SSIM ($e^{f_{\text{SSIM}}(o_t, o_t^*)}$) | 0.1/10 |
| Joint Position MSE ($e^{f_{\text{MSE}}(s_t, s_t^*)}$) | 0.1/10 |
| *Task Progress* | |
| Sub-goal Division ($\frac{n(s_t)}{|\{s^*\}|}$) | 0.1/10 |
| *Behavior Smoothness* | |
| Joint Velocity ($-|\dot{\mathbf{q}}|^2$) | 0.1/10 |
| Joint Acceleration ($-|\ddot{\mathbf{q}}|^2$) | 0.1/10 |
| Action Velocity ($-|\mathbf{a}_{t-1} - \mathbf{a}_t|^2$) | 0.01/10 |
| Action Acceleration ($-|\mathbf{a}_{t-2} - 2\mathbf{a}_{t-1} + \mathbf{a}_t|^2$) | 0.01/10 |
| *Task Completion* | |
| Task Success ($\mathbb{I}\{\text{task is successful}\}$) | 1 |

