# OpenReview forum: "RobustVLA: Robustness-Aware Reinforcement Post-Training for Vision-Language-Action Models"
_ICLR.cc/2026/Conference — Submitted to ICLR 2026_

### Official Review · Reviewer_JdVw · 2025-10-30

**Soundness:** 3
**Presentation:** 3
**Contribution:** 3
**Rating:** 4
**Confidence:** 4

**Summary:**

This paper introduces RobustVLA, a robustness-aware reinforcement post-training framework for VLA models. The authors identify two key sources of environmental disturbance—observation noise and action noise—and propose two corresponding regularizations: Jacobian regularization (to reduce input sensitivity) and smoothness regularization (to stabilize model updates).

**Strengths:**

* The paper is well-motivated and theoretically grounded, clearly linking robustness analysis with practical algorithmic design.
* The paper's focus on robust post-training of VLA models fills an important gap between pure reward maximization and real-world reliability.

**Weaknesses:**

* The experiments, though extensive, are all in simulation (LIBERO), so the claim of “real-world robustness” is not empirically validated on physical robots.
* The method’s computational overhead due to Jacobian regularization is also not clearly quantified.

**Questions:**

* How sensitive is the method to the choice of Jacobian and smoothness weights (α, β)? Is there a theoretical or empirical guideline for setting them across different tasks?
* I know this benchmark (https://github.com/sylvestf/LIBERO-plus?tab=readme-ov-file) is new, but could you test the method on this benchmark and report the performance?
* It seems like all models are added with a lot of noises during rollout, could you provide some data about what is the average noise level during rollout in reality?
* Will the loss term for enhancing robustness harm the model's general performance (i.e., the performance w/o noise)

---

> ### Author Response · Authors · 2025-11-23
> **Reply to Reviewer JdVw (Part 1)**
>
> **Q1:** The experiments, though extensive, are all in simulation (LIBERO), so the claim of “real-world robustness” is not empirically validated on physical robots.
>
> **A1:** We emphasize that our work represents a **first step** towards robust online RL post-training of VLA models, without claiming “real-world robustness” in our contributions. Please refer to **General Response 1** for our core contributions. How to perform fully autonomous, efficient, and safe online RL post-training of VLA models in the real world remains an open question. Moreover, to further verify the effectiveness of the proposed RobustVLA, we compare its performance with the state-of-the-art $\pi_{0.5}$ model on jointly perturbed LIBERO suites, as shown in the table below. The results demonstrate that our method significantly outperforms $pi_{0.5}$, endowing the model with greater robustness.
>
> | Models     | Spatial | Object | Goal | Long | Avg.         |
> |------------|---------|--------|------|------|--------------|
> | $\pi_{0.5}$    | 79.0    | 82.0   | 75.8 | 45.2 | 70.5         |
> | RobustVLA  | 88.2    | 68.2   | 83.0 | 75.2 | 78.7 **(+8.2)**  |
> | RobustVLA-C| 89.6    | 82.0   | 79.2 | 77.6 | 82.1 **(+11.6)** |
>
>
> **Q2:** The method’s computational overhead due to Jacobian regularization is also not clearly quantified.
>
> **A2:** We quantified the computational cost of Jacobian matrix regularization on four LIBERO tasks. On average, the PPO update time for the RIPT_VLA baseline is approximately 90 seconds, while our method takes approximately 100 seconds. Memory usage is very similar, mainly within the 26-27 GB range. This indicates that our method improves model robustness without consuming excessive computational resources, demonstrating computational friendliness.
>
> **Q3:** How sensitive is the method to the choice of Jacobian and smoothness weights (α, β)? Is there a theoretical or empirical guideline for setting them across different tasks?
>
> **A3:** The ablation experiments on hyperparameters $\alpha$ and $\beta$ in the paper (Figure 4(a)) show that omitting any penalty term under joint perturbation leads to a decrease in VLA model performance. The ablation results indicate that the Jacobian weight $\alpha$ and action smoothness weight $\beta$ are insensitive in the intervals [0.001, 0.02] and [0.0001, 0.002], respectively. We recommend setting the $\alpha$ and $\beta$ to approximately 0.005 and 0.0005, respectively. Theoretically, both terms play a regularization role; therefore, a moderate deviation in the weights will not fundamentally change the optimization pattern.
>
> **Q4:** I know this benchmark (https://github.com/sylvestf/LIBERO-plus?tab=readme-ov-file) is new, but could you test the method on this benchmark and report the performance?
>
> **A4:** LIBERO-plus is a relatively recent work. We attempted to fine-tune the baseline (RIPT-VLA) and our method (RobustVLA) on this benchmark; however, we were unable to successfully run the full post-training procedure due to the lack of relevant demonstration data. We will further communicate with the authors of LIBERO-plus and validate our method as soon as we obtain the relevant demonstration data. On the other hand, our theoretical analysis of observation perturbations and algorithm implementation are independent of specific forms of observation perturbations and do not make additional assumptions for specific observation perturbations. Therefore, we believe that our method has general applicability and is expected to outperform the baseline method in LIBERO-plus as well.

---

> > ### Author Response · Authors · 2025-11-23
> > **Reply to Reviewer JdVw (Part 2)**
> >
> > **Q5:** It seems like all models are added with a lot of noises during rollout, could you provide some data about what is the average noise level during rollout in reality?
> >
> > **A5:** Thank you for your constructive comments. We visualized the action distribution and added the relevant experimental results and descriptions to the paper: “We visualized the action distribution to explore how various noises affect model execution and to compare the proposed RobustVLA with the baseline RIPT-VLA. Specifically, we recorded 100 action trajectories during RIPT-VLA inference in LIBERO Spatial. Four settings were utilized: 1) no observation noise or action noise; 2) only image rotation perturbation; 3) only action noise (noise level 0.1); and 4) both image rotation perturbation and action noise. The visualization results are shown in Fig.5(a) and Appendix Fig.7. The results show that the action trajectories exhibit diverse distribution patterns and coverage under different noise conditions. This indicates that observation perturbation and action perturbation have different emphases and degrees of influence on robot manipulation behavior, especially when both perturbations are present. Furthermore, the action distribution comparison between the baseline RIPT-VLA and the proposed RobustVLA is shown in Fig.5(b) and Appendix Fig.8. The results show that under joint perturbation, the action distribution of RobustVLA is closer to the action distribution under noise-free conditions compared to RIPT-VLA. This demonstrates that RobustVLA can adaptively adjust its behavior, enabling the robot to better resist environmental disturbances and thus produce more robust and stable decision-making actions.”
> >
> > **Q6:** Will the loss term for enhancing robustness harm the model's general performance (i.e., the performance w/o noise).
> >
> > **A6:** Please refer to **General Response 2**.

---

### Official Review · Reviewer_omm7 · 2025-10-31

**Soundness:** 2
**Presentation:** 3
**Contribution:** 2
**Rating:** 4
**Confidence:** 2

**Summary:**

This paper investigates the robustness of VLA against observation space transformations and action space perturbations. The method introduces gradually adjusted noise into the training environment and proposes two regularization terms. Evaluations on LIBREO with varying noise levels show that the proposed method outperforms other baselines in success rate.

**Strengths:**

- This paper highlights the critical robustness issue in VLA models.
- The method builds on a reliable, widely adopted VLA baseline.
- The paper introduces five carefully designed observation perturbations in a common benchmark.

**Weaknesses:**

- The perturbation assumption may not be practical and seems inconsistent with the experimental setup. The state deviation is oversimplified by adding noise to the state space, and the dynamics are assumed to be Lipschitz continuous.
- There are some issues in the proof of the Theorem 1, and it appears inconsistent with the algorithm. Please refer to questions for detail.
- While the study of robustness/generalization is an established topic in visual reinforcement learning [1,2,3], adding noise to the training environment to improve evaluation performance is a well-explored approach. The specific challenges of this problem in the VLA model are not clearly addressed.

[1] Yuan Z, Ma G, Mu Y, et al. Don’t touch what matters: Task-aware lipschitz data augmentationfor visual reinforcement learning[C]// IJCAI, 2022.

[2] Fan L, Wang G, Huang D A, et al. SECANT: Self-Expert Cloning for Zero-Shot Generalization of Visual Policies[C]//ICML, 2021.

[3] Hollenstein J, Auddy S, Saveriano M, et al. Action noise in off-policy deep reinforcement learning: Impact on exploration and performance[J]. arXiv preprint arXiv:2206.03787, 2022.

**Questions:**

- In the proof of Theorem 1 (Lines 781-783), the authors use a deterministic policy by substituting the action with the policy symbol directly. However, their method employs the PPO algorithm, which is a classical stochastic policy optimization method. Could the authors clarify this difference?
- In the proof of Theorem 1 (Lines 781-782), the derivation to $\lambda \cdot \parallel \tilde{s}_t – s_t \parallel$ is unclear. Could the authors provide further clarification, and is there a risk that this results in too loose a bound?
- What is the difference between the proposed Jacobian penalty and the widely used clipped gradient norm technique?
- In the Introduction, the authors mention the out-of-distribution problem involving "unseen objects, novel environments, and different robot embodiments." However, in the experiments, they only inject simple noise into the observation and action space, along with image transformations. It would be helpful to provide a clearer and more consistent definition of the out-of-distribution scenario.

---

> ### Author Response · Authors · 2025-11-23
> **Reply to Reviewer omm7 (Part 1)**
>
> **Q1:** The perturbation assumption may not be practical and seems inconsistent with the experimental setup. The state deviation is oversimplified by adding noise to the state space, and the dynamics are assumed to be Lipschitz continuous.
>
> **A1:** We appreciate the reviewer’s concern. Our theoretical assumptions—Lipschitz-continuous dynamics and bounded additive perturbations—are indeed simplified abstractions, but they follow standard practice in the robustness and control literature. They enable an analytically tractable characterization of how observation deviations propagate through high-dimensional VLA models, without conflating these effects with environment stochasticity. Importantly, these assumptions are ***not*** inconsistent with our experiments. The visual perturbations used in LIBERO (occlusion, spatial shift, color jitter, lighting variation) and the motor noise applied during rollouts are practical realizations of bounded state and action deviations, matching the mathematical form used in our analysis. We also emphasize that our work offers **a first principled step toward robust online RL post-training for VLA models**. Compared with classical visual RL, VLA models exhibit significantly more brittle latent representations and are more prone to sharp policy drift during fine-tuning. Even under simplified assumptions, our theoretical results provide concrete guidance for designing the Jacobian and smoothness regularizers, which yield consistent gains across diverse perturbation settings. We agree that more complex disturbance models are important future directions. These extensions are conceptually compatible with our framework, but their rigorous treatment is beyond the scope of this initial robust-VLA study. Please refer to **General Response 1** for our core contributions.
>
> **Q2:** While the study of robustness/generalization is an established topic in visual reinforcement learning, adding noise to the training environment to improve evaluation performance is a well-explored approach. The specific challenges of this problem in the VLA model are not clearly addressed.
>
> **A2:**  We thank the reviewer for this important point and for the cited references. While noise injection is well-studied in visual RL, the VLA setting introduces several specific challenges that make naive noise-injection insufficient. First, VLA agents use large multimodal backbones (vision--language encoders). Their high-capacity embeddings are not merely visual features: language prompts change attention patterns, and small visual perturbations can be amplified via cross-attention into large model changes. This \emph{latent brittleness} differs qualitatively from classical CNN-based visual policies. Second, online post-training of VLA models is unstable. Even with parameter-efficient tuning (LoRA), small parameter updates can cause disproportionate output shifts (“model jitter”) because the action head depends on transformer embeddings. Simple input-level augmentation does not control this inter-iteration instability. Third, VLA tasks are often long-horizon and compositional: an early perceptual error can cascade and break higher-level task composition, so robustness must address both instantaneous perceptual sensitivity and stability of update dynamics.
> RobustVLA differs from prior augmentation-based methods in three ways: 1) ***Theory-driven decomposition:*** Theorems~1--2 separate observation sensitivity (from which we derive the Jacobian term) and accumulated degradation due to model drift (from which we derive the smoothness term). 2) ***Latent-level regularization:*** The Jacobian is computed on the final multimodal embedding (not raw pixels), targeting brittleness induced by cross-attention and language conditioning. 3) ***Training-time stability:*** The smoothness regularizer penalizes inter-iteration policy changes (and we use an adaptive noise curriculum), directly addressing the
> online-post-training instability that augmentation alone cannot fix.

---

> ### Author Response · Authors · 2025-11-23
> **Reply to Reviewer omm7 (Part 2)**
>
> **Q3:** In the proof of Theorem 1 (Lines 781-783), the authors use a deterministic policy by substituting the action with the policy symbol directly. However, their method employs the PPO algorithm, which is a classical stochastic policy optimization method. Could the authors clarify this difference?
>
> **A3:**  In the proof of Theorem~1, the notation $a_t = \pi(s_t)$ is used only as shorthand for the **conditional mean action}** $\mu_\pi(s_t) = E_{a \sim \pi}[a]$
> . Since PPO (and standard stochastic policies) optimizes the **expected return**, bounding the deviation of $\mu_\pi(s)$ is sufficient to bound the deviation in expected reward. Formally, every term of the form $\|a_t - a_t^{\*}\|$ should be interpreted as $\|\mu_\pi(s_t) - \mu_{\pi^*}(s_t)\|$,
> which satisfies the same Lipschitz/Jacobian bound used in the proof. Hence the analysis applies without loss of generality to stochastic PPO policies, and the deterministic notation is purely a notational convenience that does not restrict the result.
> Furthermore, for clarity, in the revised version of the paper, without affecting the original meaning, we have made slight modifications to the theorems, corollaries, and proofs, giving the expected version of the robust bound.
>
>
> **Q4:** In the proof of Theorem 1 (Lines 781-782), the derivation to $ \lambda \cdot \|\tilde{s}_t - s_t\|$ is unclear. Could the authors provide further clarification, and is there a risk that this results in too loose a bound?
>
> **A4:**  The step in Lines~781--782 follows a standard mean-value–type Lipschitz argument.
> For a differentiable policy mapping with bounded Jacobian
> $\|\nabla_s \pi(s)\|\le \lambda$,
>
> $$|| \pi(\tilde{s}_t) - \pi(s_t) || $$
>
> $$\le $$
> $$( \max_{u \in [s_t, \tilde{s}_t]} || \partial \pi(u) / \partial s || ) * || \tilde{s}_t - s_t || $$
>
> $$\le \lambda * \epsilon_s$$
>
> The step follows the standard Lipschitz/Jacobian bound widely used in robustness analysis of neural networks [1-3] and in stability analysis of Lipschitz dynamical systems [4-5].
> The bound is intentionally \emph{worst-case} to obtain a clean and general guarantee. In practice, the Jacobian regularizer in RobustVLA directly reduces the constant $\lambda$, which tightens the bound and correlates with the empirical robustness improvements reported in the experiments. Thus, the derivation is mathematically sound, and the resulting bound is appropriate for the purpose of establishing a tractable robustness guarantee. A more compact robust bound is in Corollary 1 and Corollary 2 in the appendix. We add the description below Theory 1 in Section 4.1: “Here the factor $L_f^{H}$ corresponds to the worst-case geometric amplification under $L_f$-Lipschitz dynamics; when $L_f<1$ the bound becomes strictly smaller.”.
>
> [1] Cisse, M., Bojanowski, P., Grave, E., Dauphin, Y., & Usunier, N. (2017, July). Parseval networks: Improving robustness to adversarial examples. In International conference on machine learning (pp. 854-863). PMLR.
>
> [2] Hein, M., & Andriushchenko, M. (2017). Formal guarantees on the robustness of a classifier against adversarial manipulation. Advances in neural information processing systems, 30.
>
> [3] Tsuzuku, Y., Sato, I., & Sugiyama, M. (2018). Lipschitz-margin training: Scalable certification of perturbation invariance for deep neural networks. Advances in neural information processing systems, 31.
>
> [4] Khalil, H. K., & Grizzle, J. W. (2002). Nonlinear systems (Vol. 3). Upper Saddle River, NJ: Prentice hall.
>
> [5] Vidyasagar, M. (2002). Nonlinear systems analysis. Society for Industrial and Applied Mathematics.
>
> **Q5:** What is the difference between the proposed Jacobian penalty and the widely used clipped gradient norm technique?
>
> **A5:** They differ fundamentally in what they constrain. Gradient clipping limits the magnitude of **parameter updates** during backpropagation to prevent numerical instability, whereas the Jacobian penalty regularizes the **model’s input--output sensitivity**, penalizing large derivatives of the model output with respect to its inputs. Thus, gradient clipping stabilizes optimization, while Jacobian regularization improves robustness to input perturbations; the two are complementary.

---

> > ### Author Response · Authors · 2025-11-23
> > **Reply to Reviewer omm7 (Part 3)**
> >
> > **Q6:** In the Introduction, the authors mention the out-of-distribution problem involving "unseen objects, novel environments, and different robot embodiments." However, in the experiments, they only inject simple noise into the observation and action space, along with image transformations. It would be helpful to provide a clearer and more consistent definition of the out-of-distribution scenario.
> >
> > **A6:** In this paper, **OOD** specifically refers to sensory-level and actuation-level distribution shifts unseen visual conditions (lighting, occlusion, camera variations) and actuation disturbances (execution noise). It does not cover large-scale domain transfer such as different robot embodiments.
> > We refined the Introduction to clearly specify this scope: “The OOD scenarios in this work primarily refer to variations in the distribution of perception and execution levels. These variations are caused by unseen visual conditions (lighting, occlusion, camera changes, etc.) and execution disturbances (action noise).”

---

### Official Review · Reviewer_taNz · 2025-10-31

**Soundness:** 3
**Presentation:** 2
**Contribution:** 3
**Rating:** 6
**Confidence:** 4

**Summary:**

This paper introduces RobustVLA, an online RL algorithm that improves the robustness of VLA models. The authors claim pre-trained VLA models fail in OOD settings due the the sensitivity to observation noise and action disturbance. To address this, they first applied theoretical analysis to bound the gap caused by noise, and then propose RobustVLA by introducing Jacobian regularization and policy smooth.

Their contributions are:

1. introduce RobustVLA, a robust RL method
2. conduct three analyses
3. experiment results show both components are effective.

**Strengths:**

1. The robustness of VLA is important.

2. Most of the idea makes sense.

3. Experiment results are good. And the authors include ablation studies.

Despite the flaws in section 4.1 (see weakness), this paper is still understandable, and the method is somewhat reasonable. I believe this paper is marginally above the acceptance threshold.

**Weaknesses:**

1. Section 4.1, the motivation to introduce the bounded Jacobian ($\left\lVert \nabla_s \pi_t(s) \right\rVert \leq \lambda$) and $\left\lVert \pi_t - \pi_{t-1} \right\rVert_\infty \leq \delta_t$ is not explained. And not all of the notations are defined.

2. Why do we need to introduce $\left\lVert \pi_t - \pi_{t-1} \right\rVert_\infty \leq \delta_t$? I believe $\left\lVert \nabla_s \pi_t(s) \right\rVert \leq \lambda$ is already sufficient to bound the gap even when actions are perturbed.

3. What is the connection between Theorem 2 and the design of $\mathcal{R}_{smooth}(\theta)$. They are telling completely different things. (1) In Theorem 2, t denotes the step in a trial. (2) But in the latter one, the comparison is between the old policy and the current policy.

Overall, these weaknesses are my major concerns. I would not mind if the paper is rejected.

**Questions:**

1. J = sum of rewards in H steps? You may want to explicitly define all the functions and variables in Section 4 (e.g. J, H, ...)
2. L437 `””` typo
3. How to interpret Figure 4 (b)(c)? All points look random.

---

> ### Author Response · Authors · 2025-11-23
> **Reply to Reviewer taNz (Part 1)**
>
> **Q1:** 1.	Section 4.1, the motivation to introduce the bounded Jacobian $\|\nabla_s \pi_t(s)\| \leq \lambda$ and $\|\pi_t - \pi_{t-1}\|_\infty \leq \delta_t$ is not explained.
>
> **A1:** Thank you for pointing this out. We revised Section~4.1: “In the perturbed online RL post-training of the VLA model, we observed two main sources of robustness degradation. First, these models are highly sensitive to small changes in visual input: slight shifts in lighting, occlusions, or camera noise can cause disproportionately large changes in the model’s latent representation and therefore in its actions. This motivates explicitly constraining how sharply the model reacts to observation changes. Second, when the model is updated online, its behavior can change abruptly from one iteration to the next. Even small gradient steps may lead to large jumps in the produced actions, which become unstable when combined with the stochasticity of realistic execution. To stabilize this process, we also need to control how quickly the model is allowed to evolve during post-training. Therefore, in this section, we theoretically quantify how different types of perturbations affect the robustness of the VLA model.”
>
> **Q2:** What is the connection between Theorem 2 and the design of R_{smooth}. They are telling completely different things. (1) In Theorem 2, t denotes the step in a trial. (2) But in the latter one, the comparison is between the old policy and the current policy.
>
> **A2:** Thank you for pointing this out. In Theorem 2 the symbol $t$ was mistakenly used to denote the model-update index, whereas it should have been the iteration index $i$. We corrected this in the revision. With this correction, the connection becomes direct. Theorem 2 contains the drift term
>
> $
> \delta_i = \|\pi_i - \pi_{i-1}\|_\infty
> $ ,
>
> which measures how much the model changes between two consecutive **training iterations**. Our smoothness regularizer,
> $R_{\text{smooth}} = E_s[ || \mu_{pi_i}(s) - \mu_{pi_{i-1}}(s) ||^2 ]$penalizes exactly this inter-iteration change. Thus, after fixing the notation, the theoretical drift term in Theorem~2 and the algorithmic design of $R_{\text{smooth}}$ are aligned: Theorem 2 shows that robustness degrades with the accumulated drift $\sum_i \delta_i$, and $R_{\text{smooth}}$
> directly controls this drift during online post-training.
>
>
> **Q3:** Why do we need to introduce
> $
> \|\pi_t - \pi_{t-1}\|_\infty \leq \delta_t
> $?I believe  $\|\nabla_s \pi_t(s)\| \leq \lambda$ is already sufficient to bound the gap even when actions are perturbed.
>
> **A3:** The Jacobian bound $\|\nabla_s \pi(s)\| \le \lambda$ regulates the model’s \emph{instantaneous sensitivity} to input perturbations. However, in online post-training, the model itself evolves over iterations. Even if a single model is smooth with respect to $s$, a large jump between two consecutive model versions can dominate the effect of the Jacobian, especially for VLA models whose outputs may shift abruptly after gradient updates. Theorem~2 shows that the return gap depends on the ***accumulated model drift*** $\sum_i \delta_i, \qquad \delta_i = \|\pi_i - \pi_{i-1}\|_\infty,$ which grows independently of the Jacobian. Intuitively, the Jacobian controls smoothness in the \emph{input dimension}, while the drift bound controls smoothness in the ***training-time dimension***. Both are necessary: even a smooth function can behave unpredictably if each update modifies it too sharply. Constraining $\delta_i$ stabilizes online adaptation and prevents compounding amplification of action noise. Moreover, we compared the performance of using Jacobian regularization and action smooth regularization under different action perturbations, as shown in the table below. The results show that the performance using action smooth regularization is significantly higher than that using Jacobian regularization. This indicates that action smoothness is indispensable for VLA models in online RL post-tracing under action perturbations.
>
> | Tasks | jacobian reg. | action smooth reg. |
> |-------|---------------|--------------------|
> | action perturbation | **0.1** | **0.2** | **0.3** | **0.1** | **0.2** | **0.3** |
> | Spatial | 94.2 | 65.0 | 29.8 | 98.2 | 73.4 | 36.8 |
> | Long | 52.0 | 15.6 | 3.8 | 77.2 | 34.6 | 7.4 |
> | Goal | 92.8 | 51.2 | 18.8 | 91.0 | 53.8 | 24.2 |
> | Object | 74.2 | 44.0 | 13.4 | 87.4 | 50.6 | 22.4 |
> | Average SR | 78.3 | 44.0 | 16.5 | 88.5 **(+10.2)** | 53.1 **(+9.1)** | 22.7 **(+6.2)** |
>
> **Q4:** J = sum of rewards in H steps? You may want to explicitly define all the functions and variables in Section 4 (e.g. J, H, ...).
>
> **A4:** Thank you for pointing this out. We added symbol descriptions to “Preliminaries”: “$J(\pi_\theta) = E\left[\sum_{t=1}^H r_t\right]$
> ,where $H$ is the rollout horizon.”. “The environmental dynamics $f(s,a)$ and the reward function $r(s,a)$ are Lipschitz continuous and are constants $L_f$ and $L_r$, respectively.”

---

> > ### Author Response · Authors · 2025-11-23
> > **Reply to Reviewer taNz (Part 2)**
> >
> > **Q5:** L437 ”” typo
> >
> > **A5:** We thank the reviewer for spotting this; it was corrected in the revision.
> >
> > **Q6:** How to interpret Figure 4 (b)(c)? All points look random.
> >
> > **A6:** Figure~4(b)(c) visualizes t-SNE embeddings of latent model representations under different visual perturbations. Although points appear scattered, RobustVLA forms compact, overlapping clusters across perturbations, whereas baselines exhibit dispersed clusters with higher variance.
> > This indicates that RobustVLA maintains more consistent latent representations under noise.

---

### Official Review · Reviewer_gru9 · 2025-11-01

**Soundness:** 3
**Presentation:** 2
**Contribution:** 2
**Rating:** 4
**Confidence:** 3

**Summary:**

This paper addresses the critical issue of Vision-Language-Action (VLA) models' sensitivity to observational noise and actuation perturbations in real-world deployments. The authors propose RobustVLA, a method that enhances model robustness by incorporating Jacobian regularization and smoothness regularization into online reinforcement learning (RL) post-training. The work is grounded in theoretical robustness analysis, which rigorously examines the impact of perturbations on performance to motivate the design of the corresponding regularizers. Extensive experiments on the LIBERO simulation platform demonstrate the method's superior performance under a variety of perturbation settings.

**Strengths:**

1. High Practical Relevance: The work astutely targets a well-known and significant vulnerability of VLA models: their fragility to observational and action perturbations in out-of-distribution (OOD) scenarios. The focus on enhancing robustness has substantial practical value for real-world robotic deployment.

2. Solid Theoretical Foundation: The theoretical analysis is a key strength. The authors establish explicit upper bounds on the performance gap under observation perturbations, action perturbations, and their combination. This provides a principled foundation and clear motivation for the proposed regularization terms.

3. Comprehensive Experimental Evaluation: The empirical validation is thorough. Experiments are conducted across diverse task suites within the LIBERO benchmark, incorporating multiple types of observation and action perturbations. The comparison against a wide array of state-of-the-art offline and online baselines makes the results compelling and convincing.

**Weaknesses:**

1. The authors note that computing the Jacobian directly on high-dimensional pixel inputs is prohibitively expensive and instead calculate it on the low-dimensional embeddings from the Llama-2 encoder used by OpenVLA-OFT. However, the paper fails to specify precisely which layer's output is used as the surrogate for the states. This lack of detail makes it difficult to fully assess the implementation's validity.

2. All experiments rely on a single, fixed pre-trained Llama-2 encoding. The work does not validate the inherent robustness of this specific encoder itself, nor does it demonstrate that the proposed method remains effective when applied to VLA models built upon different visual encoders. This raises concerns about the generalizability of the approach beyond the specific architecture tested.

3. The action perturbation is modeled exclusively as additive white Gaussian noise. Real-world robotic systems often exhibit more complex structured perturbations, such as correlated noise, latency, systematic biases and so on. The analysis and methodology do not account for these more challenging and realistic disturbance types, limiting the claimed robustness's applicability.

4. Lack of Performance-Robustness Trade-off Analysis: A critical piece of analysis is missing: the performance of RobustVLA in a completely perturbation-free, nominal environment. Without comparing its success rates to the baselines under these ideal conditions, it is impossible to evaluate the potential performance trade-off incurred by the robustness enhancements.

**Questions:**

Please see the weaknesses section

---

> ### Author Response · Authors · 2025-11-23
> **Reply to Reviewer gru9**
>
> **Q1:** The authors note that computing the Jacobian directly on high-dimensional pixel inputs is prohibitively expensive and instead calculate it on the low-dimensional embeddings from the Llama-2 encoder used by OpenVLA-OFT. However, the paper fails to specify precisely which layer's output is used as the surrogate for the states. This lack of detail makes it difficult to fully assess the implementation's validity.
>
> **A1:** In the implementation, the Jacobi regularization term is computed on the final hidden states of the Llama-2 backbone model—specifically, on the hidden states corresponding to the action tokens. This embedding is directly connected to the action head, thus capturing semantically relevant state information while ensuring the tractability of the Jacobian regularization computation.
>
> **Q2:** All experiments rely on a single, fixed pre-trained Llama-2 encoding. The work does not validate the inherent robustness of this specific encoder itself, nor does it demonstrate that the proposed method remains effective when applied to VLA models built upon different visual encoders. This raises concerns about the generalizability of the approach beyond the specific architecture tested.
>
> **A2:** Our algorithm uses a Llama-2 backbone model whose weights are updated based on the PPO loss, rather than being fixed. Our proposed method is encoder-agnostic because the regularizer acts on the final latent embedding of the backbone model and the output of the action decoder. We further conduct performance comparison experiments on the QueST+RIPT algorithm under observation perturbations, and the results are shown in the table below. Here, we apply Jacobian regularization to the state embeddings obtained from the ResNet encoder used in QueST. The results show that our method still makes the model more robust, demonstrating the generality of our approach.
>
> | Obs Perturbation (Rotation) | QueST+RIPT (baseline) | QueST+Jacobain (Ours) |
> |-----------------------------|-----------------------|-----------------------|
> | Spatial                     | 94.0                  | 96.3 (+2.3)           |
> | Long                        | 84.8                  | 85.4 (+0.6)           |
> | Goal                        | 88.5                  | 89.0 (+0.5)           |
> | Object                      | 45.0                  | 46.5 (+1.5)           |
> | Avg.                        | 78.08                 | 79.3 (+1.22)          |
>
>
> **Q3:** The action perturbation is modeled exclusively as additive white Gaussian noise. Real-world robotic systems often exhibit more complex structured perturbations, such as correlated noise, latency, systematic biases and so on. The analysis and methodology do not account for these more challenging and realistic disturbance types, limiting the claimed robustness's applicability.
>
> **A3:** We acknowledge that real-world robotic systems may be subject to more structural perturbations. Our current choice of independent, identically distributed Gaussian noise aligns with standard robustness benchmarks that are controlled and reproducibly assessable. Our primary contribution in this work is a significant step forward in robust RL post-training for VLAs. We employ a three-term robustness analysis of performance gaps introduced by observational and action perturbations, which elicit explicit regularization terms in the VLA model. RobustVLA demonstrates greater resistance to environmental uncertainties and perturbations, resulting in more reliable and stable performance. Theoretically, our analysis can be extended to correlated or biased noise: the variance term in Theorem 2 is generalized from $\sigma\sqrt{d}$ to $\sqrt{trace(\Sigma)}$ for correlated noise, while systematic bias manifests as an additive drift term. This is an important area for future work. Please refer to **General Response 1** for our core contributions.
>
> **Q4:** Lack of Performance-Robustness Trade-off Analysis: A critical piece of analysis is missing: the performance of RobustVLA in a completely perturbation-free, nominal environment.
>
> **A4:** Please refer to **General Response 2**.

---

### Author Response · Authors · 2025-11-23
**General Response**

We would like to thank the reviewers for their appreciation of our work. Reviewer gru9 recognized our work for its **high practical relevance,** **solid theoretical foundation,** and **comprehensive experimental evaluation.** Reviewer taNz praised our work as **understandable** and **reasonable.** Reviewer omm7 appreciated our work for **highlighting the critical robustness issue.** Reviewer JdVw recognized our work as **well-motivated and theoretically grounded,** and **fills an important gap**. We thank all the reviewers for their professional guidance, constructive comments, and insightful suggestions, which enabled us to improve the manuscript further. We have thoroughly performed additional experiments and revised the manuscript to address all your comments and concerns. The major revisions in the manuscript are marked in blue.

## **General Response:**

## 1.The core contribution of this work.

(1) We reveal a **core limitation** of existing RL-based VLA post-training—they optimize reward but **do not** control robustness against observation or action perturbations.

(2) We provide a **principled robustness analysis** that yields two explicit regularizers: Jacobian regularization for observation noise; Smoothness regularization for action perturbations. This theoretical grounding is absent in prior work.

(3) We propose **RobustVLA**, a lightweight online RL post-training method that incorporates these regularizers to explicitly enhance robustness.

(4) We demonstrate **consistent, significant robustness gains** across diverse perturbation settings, surpassing state-of-the-art VLA baselines.

## 2.Performance-Robustness Trade-off Analysis.

We have performed a nominal (unperturbed) evaluation on the LIBERO tasks. The results show that RobustVLA's success rate is almost identical to RIPT-VLA (**the average performance difference is only 1.15.**). This shows that although our regularization term is proposed to address how the model responds to external disturbances, it does not drastically reduce the model's performance in the undisturbed environment.

| Perturbation-free Env. | RIPT-VLA (baseline) | RobustVLA (Ours, using jacobian+smooth reg.) |
|------------------------|---------------------|----------------------------------------------|
| Spatial                | 98.2                | 98.4 (+0.2)                                  |
| Goal                   | 98.4                | 99.0 (+0.6)                                  |
| Long                   | 93.8                | 92.4 (-1.4)                                  |
| Object                 | 98.6                | 94.6 (-4.0)                                  |
| Average SR             | 97.25               | 96.1 (-1.15)                                 |

## **Paper Revision Updates:**

We have submitted a revised version of the paper, with the main changes highlighted in blue. The following is the updated content:

1. The INTRODUCTION section has modified the OOD scenarios for these works.

2. The PRELIMINARIES section has added explanations for related symbols such as $J_{\pi_{\theta}}, H, L_f, L_r$.

3. The motivation for introducing the Jacobian and motion smoothness concepts has been added to section 4.1 of METHODOLOGY.

4. The relevant theorems and corollaries in section 4.1 of METHODOLOGY and the appendix have been revised to the expected versions. The relevant symbols have been explicitly explained.

5. In section 5.2 of EXPERIMENTAL EVALUATION, the relevant task names have been changed to italics.

6. We have added experimental results related to the action distribution in section 5.3 ABLATION ANALYSIS and the appendix.

Again, we sincerely thank the reviewers for their constructive feedback. We believe all comments have been addressed in this revision but are happy to address any further comments from the reviewers.

---

### Author Response · Authors · 2025-11-27
**Friendly Reminder Regarding Upcoming Rebuttal Deadline**

Dear reviewers,
I hope this message finds you well.

This is a gentle reminder regarding the review of our manuscript. We deeply appreciate the invaluable comments and feedback provided by reviewers. They are instrumental in enhancing the quality of our research. As per the schedule, the rebuttal phase is drawing to a close. We understand that you have a demanding schedule and a multitude of responsibilities, but we are keen to receive your feedback before the deadline. This will afford us the opportunity to address any questions or concerns you may have raised in a timely manner. We are eager to incorporate your insights to refine our work and would be grateful if you could share your thoughts prior to the rebuttal deadline.

Thank you very much for your hard work and support. Your dedication to the review process is greatly appreciated.

---

### Meta-Review · Area_Chair_xHRp · 2026-01-07

**Summary:**

The paper proposes RobustVLA, a method to enhance the robustness of Vision-Language-Action models against observational and action perturbations through online RL post-training. Specifically, the authors introduce Jacobian and Smoothness regularization terms.

While the reviewers acknowledged the importance of the problem and the theoretical motivation, the scores were  4, 6, 4, 4. Despite a detailed rebuttal that addressed specific implementation questions, fundamental concerns regarding the gap between the theoretical assumptions and real-world applicability, as well as the lack of physical validation, remain unresolved.
My recommendation is reject, and the decision is driven by the following three primary factors:
1. Lack of Real-World Validation (Reviewer JdVw):
The central claim of the paper is Robustness for VLA models, which are inherently designed for robotic manipulation. However, the experiments are conducted on the LIBERO simulator. Reviewers JdVw strongly emphasized that "real-world robustness" cannot be claimed without real-world experiments.
2. Oversimplified Noise & Assumptions (Reviewer gru9, Reviewer omm7):
Reviewers gru9 and Reviewer omm7 pointed out that the method relies on additive white Gaussian noise and Lipschitz continuity assumptions. Real-world robotic system noises exhibit more structured perturbations (e.g., latency, systematic bias).
3. Incremental Contribution (Reviewer omm7):
Reviewer omm7 noted that adding noise to training to improve evaluation performance is a well-explored technique, and the specific novelty for VLAs beyond standard regularization was not convincingly argued.

**Reviewer Concerns:**

- Concerns Addressed by Rebuttal
  - Implementation Details (Reviewer gru9, Reviewer JdVw): Clarified that the Jacobian is computed on the final hidden states and provided computational cost metrics.
  - Applicability to Different Encoders (Reviewer gru9): The authors validated the effectiveness of the proposed method on the QueST encoder, proving the method isn't tied to Llama-2.
  - Theoretical Confusion (Reviewer taNz): The authors corrected the notation errors and explained the distinct roles of Jacobian and Smoothness regularization terms, which satisfied Reviewer taNz's technical questions.
- Outstanding Concerns
  - Simulation-Only Scope: The rebuttal confirmed that no real-robot experiments were performed. The authors' comparison to pi0.5 is still within LIBERO. This fails to address the core concern of Reviewer JdVw regarding "real-world robustness."
  - Real-world Robotic Perturbations: The authors acknowledged that real-world noise is structural but maintained their Gaussian assumptions for tractability. This leaves the method's practical utility unproven against the complex disturbances mentioned by Reviewer gru9.

**Reviewer Scores:**

Reviewer gru9 would likely maintain a score of 4, as the fundamental critique regarding the gap between simple Gaussian noise models and complex real-world structured perturbations remains unaddressed.

Reviewer taNz would likely keep their score at 6.  Although Reviewer taNz acknowledges the theoretical clarifications, he explicitly states that "would not mind if the paper is rejected".

Reviewer omm7 is expected to hold at 4, as the rebuttal defended the theoretical assumptions but failed to empirically resolve the concerns regarding the practicality of the perturbation models.

Reviewer JdVw would likely remain at 4, because the rebuttal confirmed the absence of real-world experiments, which are critical for validating the paper's claim of 'real-world robustness.

---

### Decision · Program_Chairs · 2026-01-26

Reject